# ATR, a DNA damage kinase, modulates DNA replication timing in *Leishmania major*

**Gabriel L. A. da Silva** [1]*, **Jeziel D. Damasceno**[1], **Jennifer A. Black**[2], **Craig Lapsley**[1], **Richard McCulloch**[1]*, **Luiz R. O. Tosi**[2]*

**1** University of Glasgow Centre for Parasitology, The Wellcome Centre for Integrative Parasitology, University of Glasgow, School of Infection and Immunity, Glasgow, United Kingdom, **2** Department of Cell and Molecular Biology, Ribeirao Preto Medical School, University of Sao Paulo, Ribeirao Preto, Brazil

\* gabriel.almediadasilva@glasgow.ac.uk (GLAdS); luiztosi@fmrp.usp.br (LROT); Richard.mcculloch@glasgow.ac.uk (RMC)

## Abstract

All cells possess mechanisms to maintain and replicate their genomes, whose integrity and transmission are constantly challenged by DNA damage and replication impediments. In eukaryotes, the protein kinase Ataxia-Telangiectasia and Rad3-related (ATR), a member of the phosphatidylinositol 3-kinase-like family, acts as a master regulator of the eukaryotic response to DNA injuries, ensuring DNA replication completion and genome stability. Here we aimed to investigate the functional relevance of the ATR homolog in the DNA metabolism of *Leishmania major*, a protozoan parasite with a remarkably plastic genome. CRISPR/cas9 genome editing was used to generate a Myc-tagged ATR cell line (mycATR), and a Myc-tagged C-terminal knockout of ATR (mycATRΔC-/-). We show that the nuclear localisation of ATR depends upon its C-terminus. Moreover, its deletion results in single-stranded DNA accumulation, impaired cell cycle control, increased levels of DNA damage, and delayed DNA replication re-start after replication stress. In addition, we show that ATR plays a key role in maintaining *L. major's* unusual DNA replication program, where larger chromosomes duplicate later than smaller chromosomes. Our data reveals loss of the ATR C-terminus promotes the accumulation of DNA replication signal around replicative stress fragile sites, which are enriched in larger chromosomes. Finally, we show that these alterations to the DNA replication program promote chromosome instability. In summary, our work shows that ATR acts to modulate DNA replication timing, limiting the plasticity of the *Leishmania* genome.

## Author summary

A series of protein kinases act at the pinnacle of eukaryotic cells' DNA damage response, organising and guiding repair. One such kinase is ATR, which recognises single-stranded DNA formed during DNA replication errors and DNA break

**Data availability statement:** The data for this study have been deposited in the European Nucleotide Archive (ENA) at EMBL-EBI under accession number PRJEB83661. Data and script underlying the main and supplementary figures are provided in S1-S3 Files.

**Funding:** This work was supported by FAPESP [16/16454-9, 19/20731-6 to LT], the Wellcome Trust [224501/Z/21/Z to RM], and the BBSRC [BB/N016165/1, BB/R017166/1, BB/W001101/1 to RM]. The funders had no role in study design, data collection and analysis, decision to publish, or preparation of the manuscript.

**Competing interests:** The authors have declared that no competing interests exist.

repair. How ATR acts in the important protozoan parasite *Leishmania* remains unclear, and so we generated a gene knockout mutant that revealed a series of changes in cell behaviour consistent with known ATR functions. Additionally, our ATR mutant unexpectedly altered the pattern of chromosome replication timing, revealing a wider role in the organisation of how genome duplication is programmed. Such a change in DNA replication programming is associated with increased genome content variation.

## Introduction

DNA stability is constantly challenged by a range of stressors that can cause a diversity of DNA lesions [1,2]. To tackle all possible lesion types and guarantee genome integrity, cells have evolved a range of pathways, collectively known as the DNA Damage Response (DDR), which when activated lead to damage recognition, signalling and ultimately repair [2–4]. At the pinnacle of the eukaryotic DDR are three protein kinases that act to recognise and signal DNA damage, triggering the activation of the appropriate repair response: ATM (Ataxia-Telangiectasia Mutated) and DNA-PKcs (DNA protein kinase catalytic subunit) are recruited primarily to DNA double strand breaks (DSBs), whereas ATR (Ataxia-Telangiectasia and Rad3-related) primarily responds to the accumulation of single-stranded DNA (ssDNA) during DNA replication [3–5].

Stalling or slowing of the replisome leads to ssDNA accumulations during DNA replication [6]. Exposed ssDNA is protected by the RPA complex, limiting degradation by nucleases [6,7]. The resulting ssDNA-RPA complex, along with 5'-ended ssDNA-dsDNA junctions formed at stalled replication forks, serves as a platform for the recruitment of ATR-interacting protein (ATRIP), TOPBP1 (Topoisomerase II-binding protein 1) and the heterotrimeric 9-1-1 checkpoint complex (Rad9-Rad1-Hus1) [8–10]. ATRIP recruits ATR to the damage site, while TOPBP1 and 9-1-1 complex are responsible for the stabilisation and full activation of ATR [10–13]. In addition to these factors, in some organisms ETAA1 also activates ATR by directly interacting with RPA [14,15]. Once activated, ATR coordinates a complex response that safeguards the replication fork to preserve DNA integrity [14]. Instrumental to this response is the effector kinase CHK1, which is activated upon phosphorylation by ATR [16,17]. ATR-driven CHK1 activation functions to arrest the cell cycle, suppress origin firing, stabilises replication forks, and promotes fork repair and DNA replication restart [18,19]. By coordinating cell cycle arrest with fork stabilization and recovery, ATR and CHK1 prevent cells from entering mitosis when DNA replication is compromised [19,20].

Most of our understanding of ATR's- activities comes from studying mammalian and yeast cells [21–24]. Far less research has examined how the ATR pathway operates in eukaryotic pathogens [25]. *Trypanosoma brucei*, *Trypanosoma cruzi* and *Leishmania spp.* are related protozoan parasites, part of a wider grouping known as trypanosomatids. These protozoa cause vector-bone diseases affecting millions of people every year in tropical and sub-tropical regions of the globe, though recent

cases of Chagas disease and leishmaniasis are emerging across Europe and North America [26–29]. All three pathogens are notable for complex adaptive changes to their genome composition to survive different environments throughout their life cycles [30–32]. For instance, *T. brucei* and *T. cruzi* display elevated levels of recombination among large, variable gene families that encode cell surface proteins required for evasion of the host immune system [33–35]. *Leishmania* lack such variable gene families, but genome plasticity is remarkably widespread, with both natural isolates and laboratory strains showing chromosomal aneuploidy and gene copy number variation (CNV). Such alterations to their genome composition are commonly associated with drug resistance emergence and environmental adaptations [31,32,36–38].

Several studies have linked DDR catalytic factors to mutation and Copy Number Variation (CNV) [39–41], but fewer have examined how damage signalling may influence genome stability [25]. To date, there has been no investigation into whether or not *Leishmania* DDR kinases modulate genome stability. Studies in *T. brucei* show that ATR loss has a distinct impact depending upon the life cycle stage: in insect procyclic stages, RNAi-mediated depletion of ATR leads to a modest reduction in cell proliferation and cell cycle progression following ionizing radiation exposure [42], while in mammalian-infective cells ATR depletion is lethal and disrupts VSG expression [42,43]. In *Leishmania*, initial studies investigating the potential functions of ATR used human-specific ATR inhibitors, reporting a reduction in cell proliferation, increased sensitivity to oxidative stress, and an accumulation of ssDNA [44,45]. Gene knockout experiments in *L. mexicana* suggested that ATR is not essential for cell survival *in vitro* [46], and our recent work suggests that ATR deficiency in *L. major* exacerbates replication stress [47]. In this study, we used CRISPR/Cas9 genome editing to generate an ATR mutant cell line. Although a complete KO of the *ATR* gene could not be achieved in *L. major*, we show that loss of the predicted kinase domain, located within the C-terminus of the kinase, disrupts nuclear localization and is tolerated. Using this mutant, we demonstrate that ATR is involved in regulating the unusual chromosome size-dependent timing of *Leishmania* DNA replication, and that disrupting this process leads to compromised genome stability and variation.

## Results

### Generation of ATR mutants lacking the C-terminal kinase domain in *Leishmania major*

To investigate the roles of ATR in *L. major*, we sought to genetically modify the *ATR* locus (LmjF.32.1460) for two purposes. To first localise the protein, we used CRISPR-Cas9 to generate cells expressing full length ATR tagged at its N-terminus by inserting mNeonGreen (mNG) and three copies of the Myc tag [48,49] upstream of the *ATR* ORF (Fig 1A); these cells are referred to as mycATR. Second, to assess ATR function, we aimed to delete the *ATR* ORF. However, all attempts to remove the entire *ATR* ORF using CRISPR/Cas9 were unsuccessful, suggesting ATR is essential in promastigote forms of *L. major*. Therefore, in our mycATR cells we opted to delete a 3,444 bp region from the C-terminus of *ATR* (6218 – 9624 bp), containing a predicted FAT domain, catalytic domain (CD) and FATC domain, generating the cell line mycATRΔC-/- (Fig 1A).

PCR amplification of the N- or C-termini of the *ATR* locus revealed that a single round of antibiotic selection was successful to tag the N-terminus of *ATR* and to delete the C-terminal coding domain of both alleles (Fig 1B). We independently generated two such deletion clones, cl1 and cl8. Additionally, we recovered a clone in which only one allele of *ATR* was truncated, referred to as mycATRΔC+/-. Expression of tagged ATR in the mycATR and mycATRΔC-/- cells was assessed by western blotting, revealing bands of the expected size: full-length tagged ATR migrated just below a 460 kDa marker (predicted size, 383.31 kDa), while the C-terminally truncated version appeared at approximately 260 kDa (predicted size, 256.42 KDa; Fig 1C). Quantification of myc signal on the blot further showed that the C-terminal deletion reduced total ATR protein levels, indicating that this region could be important for ATR stabilisation and/or expression (Fig 1C).

Next, we examined the growth profile of mycATR, mycATRΔC-/-, and mycATRΔC+/- cell lines (Fig 1D), comparing their growth to that of parental *L. major* CC1 and Cas9-T7 cells (S1A Fig). The results showed that loss of one *ATR* allele or N-terminal tagging of full-length ATR did not affect population doubling (Figs 1D and S1A). In contrast, proliferation of the

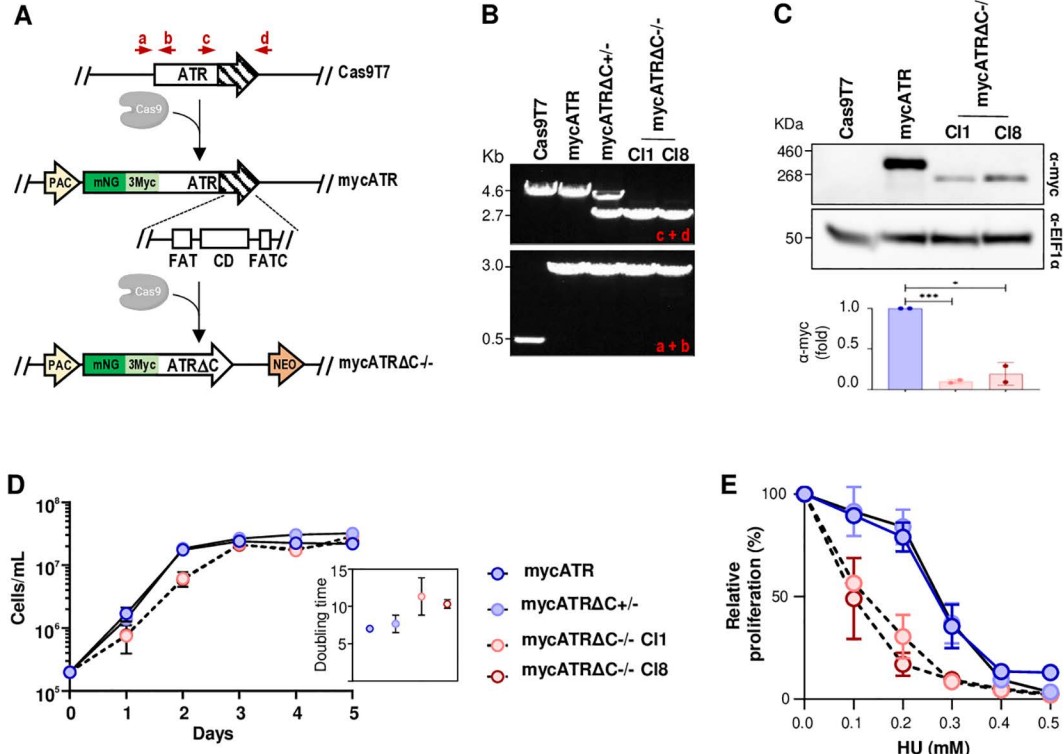

**Fig 1. Deletion of the C-terminus of *L. major* ATR alters the protein's abundance and sensitivity to replication stress.** (A) Schematic representation showing a cell line expressing Cas9, which was used to add a tag (3xmyc + NeonGreen) N-terminally of the ATR kinase (LmjF.32.1460), generating mycATR cells. The predicted kinase domain present in the deleted C-terminal region (6218 – 9624 bp) is highlighted as a dashed area which contains a predicted FAT domain, catalytic domain (CD) and FATC domain. The C-terminal of the mycATR cell line was replaced by a selectable marker (NEO), generating the mycATRΔC-/- cell line; red arrows represent primers used to detect these alterations. (B) PCR analysis from mycATR, mycATRΔC + /- and mycATRΔC-/- cells, indicating the addition of the tags in two alleles of both cells (a + b) and the deletion of the C-terminal region (c + d) in both alleles of mycATRΔC-/- cells. (C) Western analysis of whole cell extracts of indicated cells in exponential growth; extracts were probed with α-myc antiserum, and α-EIF1α was used as a loading control (predicted protein sizes are indicated, kDa). Quantification of the bands from the western blot using ImageJ software; statistical analysis was performed using Prism software (Error bars ± SD; n = 2 experiments, * p < 0.05; ** p < 0.005; *** p < 0.001; **** p < 0.0001. Unpaired t-test). (D) Growth curve of the indicated cells cultivated in HOMEM medium; cells were seeded at ~2x10⁵ cells.ml⁻¹ at day 0; growth was evaluated over 5 days and cell density was assessed every 24 hours (Error bars ± SD; n = 2 experiments). The doubling time (small box) was calculated using the exponential growth coefficient (GC) from day 1 to day 3, applied by the following equation: =(LN(2)/(GC))*24. (E) Cell survival curve of indicated cells cultivated in HOMEM medium; cells were seeded at ~2x10⁵ cells.ml⁻¹ and left untreated or treated with the indicated concentrations of Hydroxyurea HU (mM); after 96 hours cell density was assessed, and the percentage was calculated relative to the density of untreated cells.

mycATRΔC-/- cells was reduced compared to mycATR and mycATRΔC + /- cells, resulting in an increase in doubling time (mycATRΔC-/- clones, ~ 11 and 10 hours, versus ~7 and 8 hours for mycATR and mycATRΔC + /- cells, respectively; Fig 1D). Despite this difference in proliferation, FACS analysis of DNA synthesis, assessed by IdU incorporation, showed similar percentages of replicating cells in the mycATR and mycATRΔC-/- populations (S1B). Also, all mutant clones reached the same maximum cell density as the mycATR and mycATRΔC + /- cells (Fig 1D). We next assessed the sensitivity of mycATR, mycATRΔC + /- and mycATRΔC-/- cells to hydroxyurea (HU), a reversible inhibitor of ribonucleotide reductase. Cells were treated with increasing concentrations of HU or left untreated. After 96 hours of growth, we measured cell density in the HU-treated populations relative to untreated controls. The results showed that around 50% of mycATRΔC-/- cells grew slower or died more quickly in the presence of 0.1 mM HU compared with 2% from both mycATR and mycATRΔC + /- cells (Fig 1E). Overall, these findings indicate that ATR C-terminal deletion alters promastigote growth dynamics and increases sensitivity to HU, highlighting a role for ATR during the response to replication stress in *L. major*.

### The C-terminus of ATR is vital for accumulation of the protein in the nucleus

In other eukaryotes, including *T. brucei*, ATR has been shown to localize to the nucleus [43,50,51]. However, the subcellular localization of ATR in *Leishmania* species remains unknown from available data (Leishtag.org). To investigate the subcellular location of ATR in *L. major* promastigotes and ask if deletion of the C-terminal plays a role in ATR localisation, we performed indirect immunofluorescence (IFA) on formaldehyde-fixed mycATR and mycATRΔC-/- cells using anti-myc antiserum (α-myc). Images were captured using structured illumination super-resolution microscopy (SR-SIM), allowing for high-resolution visualization of ATR localization (Fig 2A). IFA revealed accumulation of α-myc signal within the nucleus of mycATR cells. In contrast, mycATRΔC-/- cells showed little nuclear localization, with α-myc signal distributed throughout the cell body, including signal throughout the cytoplasm (Fig 2A). Quantification of the nuclear myc signal confirmed a significant decrease in nuclear myc signal intensity in mycATRΔC-/- cells compared with mycATR cells (Fig 2B), suggesting that deletion of the C-terminus disrupts nuclear localisation of ATR. In support of a nuclear localisation for ATR directed by the C-terminus, we identified a putative nuclear location signal (NLS) within the deleted region of the *ATR* ORF (amino acids 2922–2932) (S2A Fig).

To further validate the subcellular location of ATR, we performed live cell microscopy using mycATR cells, but this time detecting mNG signal (S2B Fig). Additionally, we tagged ATR with just three copies of Myc (3MATR) to ask if mNG tagging altered protein localisation. Both approaches confirmed the accumulation of ATR within the nucleus (S2B and S2C Fig). Altogether, we suggest that the growth and HU sensitivity phenotypes associated with the loss of the C-terminus of ATR may arise from a combination of lost C-terminal functions, reduced protein abundance and impaired accumulation in the nucleus.

### Replication stress leads to ssDNA accumulation, which affects the mechanisms influencing cell cycle progression in mycATRΔC-/- cells

The primary substrate of ATR activation is elongated tracks of ssDNA [13,52]. In ATR's absence, ssDNA accumulates in human cells under replication stress conditions [50,53]. Given that even partial deficiency of ATR increases ssDNA levels in *L. major* [47], we investigated whether deletion of the C-terminus of ATR leads to ssDNA accumulation. To assess this, we labelled DNA by growing mycATR and mycATRΔC-/- cells in medium containing 150 µM IdU, a thymidine analogue, for 16 hours (around two S phases). Cells were then washed and cultivated in the presence or absence of 5 mM of HU for 5 hours. Cells were fixed and stained with α-BrdU antiserum to detect the incorporated IdU, with α-BrdU signal reflecting accumulation of ssDNA, since DNA was not denatured. Microscopy images showed a clear increase of nuclear IdU signal in HU-treated mycATRΔC-/- cells compared with untreated (Fig 3A). In addition, quantification of the signal intensity revealed a significant increase in the both HU-treated and -untreated mycATRΔC-/- cells in comparison with mycATR cells (Fig 3B), indicating that loss of ATR leads to the accumulation of ssDNA in *L. major*.

Considering that one of the primary functions of ATR, once activated by ssDNA accumulation, is cell cycle regulation [54–56], we next examined the cell cycle profiles of mycATR and mycATRΔC-/- cells by flow cytometry with and without HU exposure in exponentially growing cells. We selected two different conditions to examine the effect of ATR C-terminal loss: chronic (0.5 mM of HU for 20 h) or acute replication stress challenge (5 mM of HU for 5 h). Cells were collected and fixed after being cultured in fresh media for 0, 2 and 5 h following chronic HU treatment release, or for 0, 2 and 4 h after acute HU treatment. The FACS profile of propidium iodide-stained DNA of mycATR cells showed that chronic treatment did not completely block cell cycle progression; rather, the progression through S phase was slowed. Specifically, there was an initial accumulation of cells in early S phase at 0 h, followed by an accumulation in mid/late S phase at 2 h after HU release. By 5 hours, the cell cycle profile had returned to levels like those seen in untreated cells (Fig 3C). The higher concentration of HU during the acute treatment induced a cell cycle stall in the mycATR cells at the G1/S boundary (0 h, Fig 3C). After release from HU, mycATR cells progressed through S phase into G2/M at 2 and 4 h, respectively (Fig 3C). In contrast, both HU treatments had a distinct effect on cell cycle progression in mycATRΔC-/- cells. After chronic HU

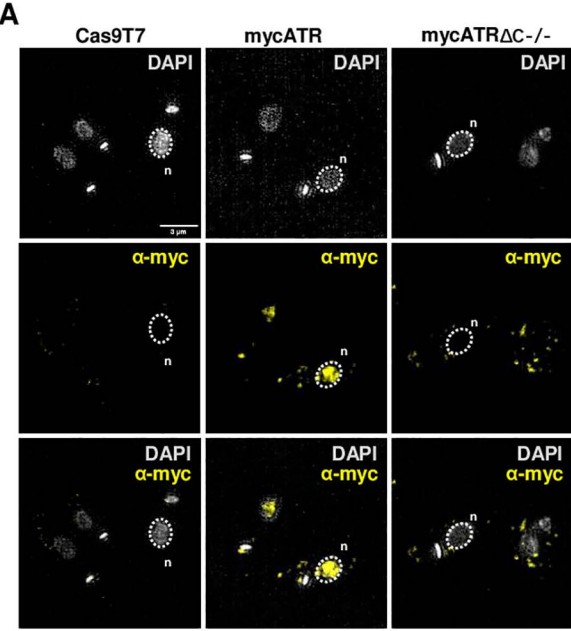

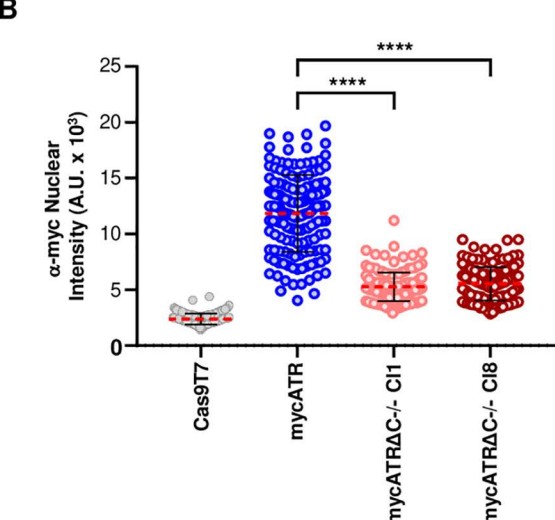

**Fig 2. Subcellular localisation of *L. major* ATR kinase depends on its C-terminus.** (A) Representative images of the sub-cellular localization of myc signal, which is added N-terminally to ATR in mycATR and mycATRΔC-/- cells; scale bar = 3 μm. Images were captured on an Elyra super resolution microscope (Zeiss). Nuclear (n) and Kinetoplast (k) DNA are shown in gray; and the myc signal in yellow. (B) Quantification of the myc signal around the nucleus. An area was drawn around ≥ 100 nuclei based on DAPI staining and myc signal was measured using ImageJ software, Cas9T7 cells were use as negative controls (Error bars ± SD; n = 2 experiments, **** P < 0.0001. Unpaired t-test).

exposure, we detected an increased accumulation of mycATRΔC-/- cells in S phase at 0 h (Fig 3C), which did not obviously alter at either 2 or 5 h following HU release. Moreover, we observed an increase in the number of sub-G1 cells (<G1) following chronic HU exposure in the absence of the C-terminus of ATR (Fig 3C). After acute HU treatment, mycATRΔC-/- cells synchronised at the G1/S boundary, as observed in mycATR cells (0 h, Fig 3C) and progressed through S phase to G2/M between 2 and 4 h, but with a substantial rise in the <G1 population (Fig 3C). Taken together, these data suggest

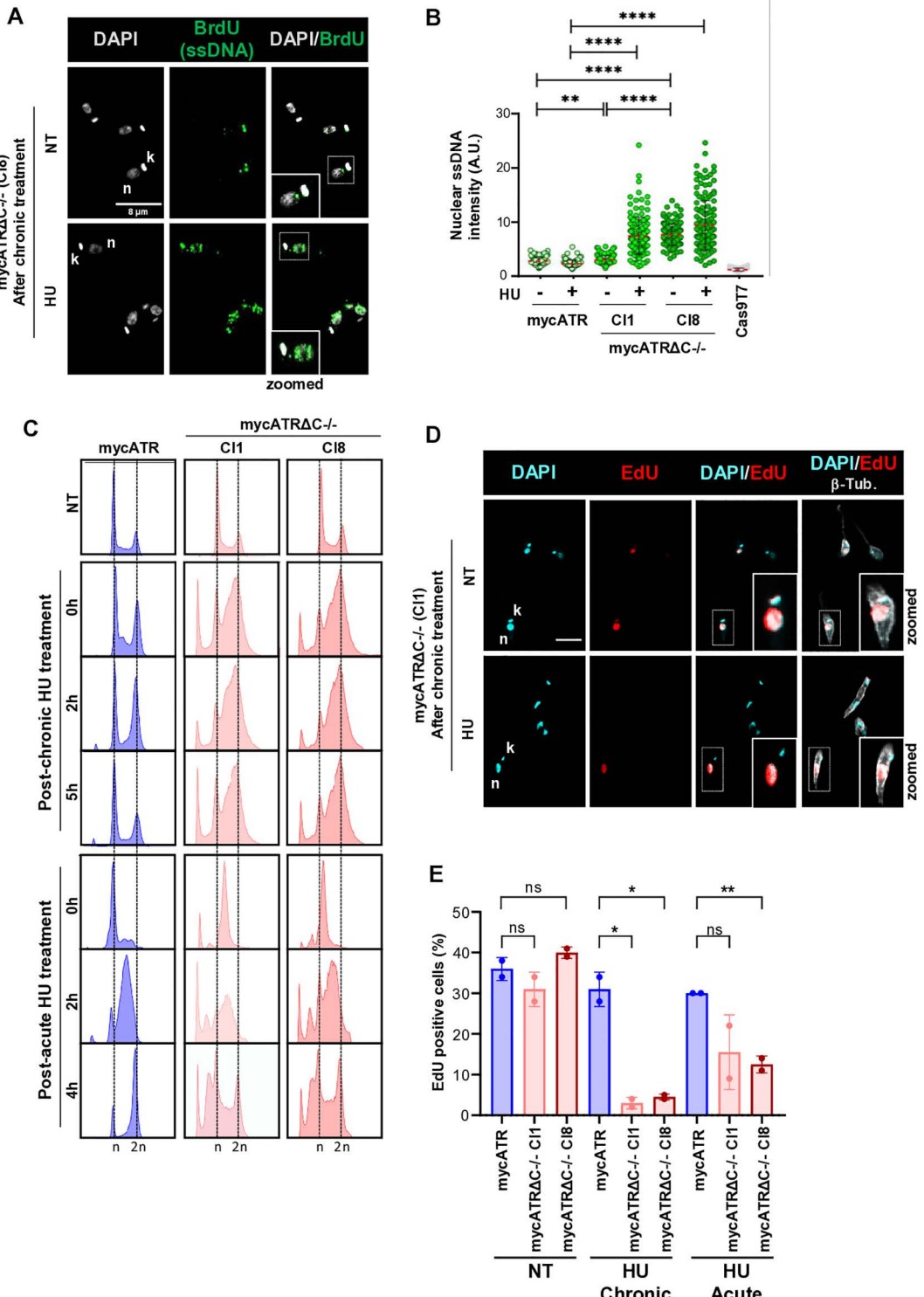

**Fig 3. Replication stress leads to ssDNA accumulation, which affects the mechanisms influencing cell cycle progression in mycATRΔC-/- cells.** (A) Representative images acquired by a LSM880 Zeiss confocal microscope from Z stack images of mycATRΔC-/- cells (cl8) untreated (NT) or treated with 5 mM of HU for 5 hours. Previously, cells were incubated with 150 μM of IdU for 16 hours for DNA labelling. Cells were fixed with 3% formaldehyde and stained with α-BrdU (ssDNA, green); genomic DNA was stained using DAPI (gray). Dashed squared delimitated a cell zoomed out. (n)

nucleus (k) kinetoplast are indicated. Bar scale = 8 um. (B) Quantification of the ssDNA signal in the nucleus of mycATR and mycATRΔC-/- cells treated or not with 5 mM of HU for 5 hours; Cas9T7 cells without α-BrdU were use as negative control. An area was drawn around 150 nuclei based on DAPI staining and ssDNA signal was measured using ImageJ software (Error bars ± SD; n = 2 experiments, p < 0.05; ** p < 0.005; *** p < 0.001; **** p < 0.0001. Unpaired t-test). (C) mycATR and mycATRΔC-/- cells were left untreated (NT) or treated with 0.5 mM of Hydroxyurea (HU) for 20 hours (Chronic) or treated with 5 mM of Hydroxyurea (HU) for 5 hours (Acute), washed and resuspended in fresh media, and samples collected at 0, 2, 5 hours after chronic treatment and 0, 2, 4 hours after acute treatment. Cells were fixed with methanol and DNA stained with propidium iodate, followed by flow cytometer analysis; the cell cycle profile was created using FlowJo software. (D) Representative images of mycATRΔC-/- (cl1) cells pulsed with 150 µM of EdU for 30 minutes in untreated condition or after 5 hours culturing in fresh media post release from chronic HU treatment. Cells were fixed with 3% formaldehyde and a Click-IT reaction was used to stain cells that had uptaken EdU during the 30 min pulse (red); genomic DNA was stained with DAPI (cyan) and α-tubulin was used to stain the cytoskeleton (gray). Dashed squared delimitated a cell zoomed out. (n) nucleus (k) kinetoplast are indicated. Bar = 6 um. (E) Percentage of EdU positive mycATR or mycATRΔC-/- cells in each condition (untreated, after release from chronic (post 5 hours), or acute (post 4 hours) HU treatment) was calculated by the number of positive EdU stained cells divided by the total number of cells present in each sample (100 – 300 cells; error bars ± SD; n = 2 experiments, * p < 0.05; ** p < 0.005; *** p < 0.001; **** p < 0.0001. Unpaired t-test).

that deleting the C-terminus of ATR leads to cell cycle defects after HU exposure. Chronic treatment with 0.5 mM HU impairs cell cycle progression of mutant cells once they reach S phase and induces cells with aberrant DNA content, while acute treatment with 5 mM HU leads to cell cycle G1/S synchronisation, but a compromised progression after HU release, which it is also associated with the generation of pronounced levels of aberrant cells. Overall, mycATRΔC-/- cells leads to aberrant cell cycle progression under different levels of DNA replication stress.

The ability of stalled DNA polymerases to restart DNA synthesis after removal or bypass of a block in DNA replication elongation is an indication of replication fork stabilization [57–59]. To further examine the effects of replication stress in the absence of ATR's C-terminus, we investigated the percentage of mycATR and mycATRΔC-/- cells capable of restarting DNA synthesis after replication stress. Untreated cells, cells 5 h post-release from chronic HU exposure, or 4 h after release from G1/S arrest by acute HU exposure were pulsed with 150 uM EdU for 30 min. Cells were then fixed and a Click-IT reaction was performed to detect cells that incorporate the nucleotide analogue (as a proxy for DNA synthesis; Fig 3D). The percentage of EdU positive cells were identified by microscopy and calculated as a proportion of the total of cells population (Fig 3E). In untreated conditions, we detected no difference in the percentage of EdU-positive mycATR and mycATRΔC-/- cells (~30% for both; Fig 3E). On the other hand, following either chronic or acute HU treatment, the percentage of EdU-positive mycATRΔC-/- cells was significantly reduced compared with mycATR cells, especially after HU chronic treatment (Fig 3E). In addition, the nuclear signal of EdU positive cells also showed an overall difference between mycATR and mycATRΔC-/- cells, suggesting that mechanisms involving DNA synthesis, such as DNA replication, DNA repair and DNA recombination, are affected after replication stress in cells with mutated ATR (S3 Fig).

## ATR C-terminal deletion results in nuclear DNA damage

Our results above suggest that loss of ATR's C-terminus coincides with cell cycle defects and enhanced genome instability. To test this interpretation, we next examined the levels of γH2A, a DNA damage marker in trypanosomatids [60]. mycATR and mycATRΔC-/- cells were harvested in the absence of replication stress or following exposure to chronic and acute HU treatment as described above. After treatment, whole cell extracts were prepared at 0 and 5 h post-release chronic treatment, and 0 and 4 h post-release acute treatment and the levels of γH2A assessed by western blotting. γH2A signal was detected using anti-T. brucei γH2A antiserum that cross reacts with Leishmania γH2A [60]. γH2A signal was quantified for all cell lines and intensity calculated relative to untreated mycATR cells (Fig 4A–4D). After chronic HU treatment, a significant increase in γH2A levels was observed in mycATRΔC-/- cells when compared with mycATR cells (Fig 4A and 4C). However, after acute HU treatment, γH2A levels were similar in both cell lines (Fig 4B and 4D). These findings suggest that the loss of the C-terminus of ATR results in enhanced genome damage when L. major cells are able to sustain DNA replication under chronic HU treatment. Furthermore, the G1/S synchronization observed after acute HU treatment caused increased DNA damage in mycATR cells, a level of damage that is not further aggravated in mycATRΔC-/- cells.

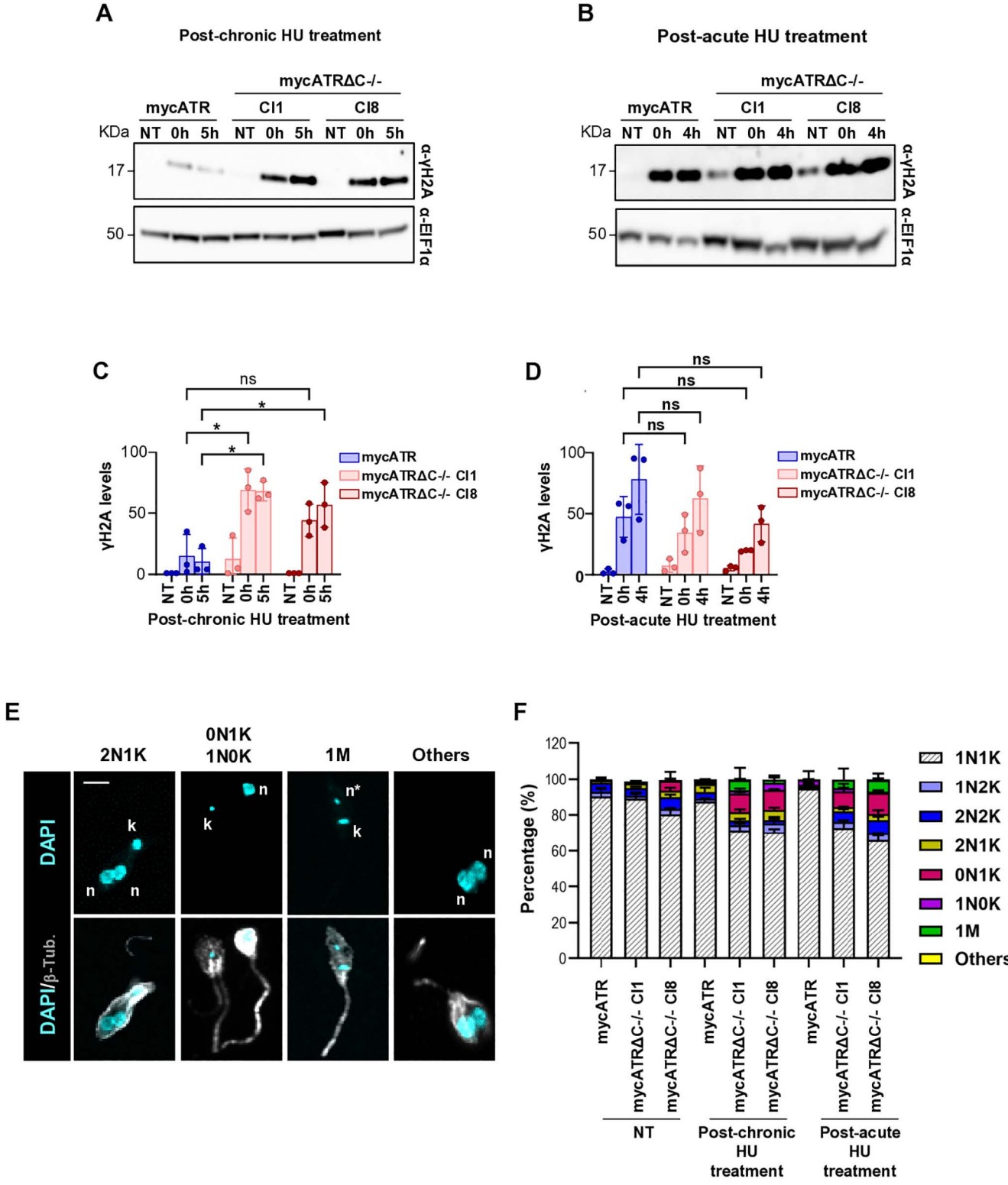

**Fig 4. Deletion of the C-terminus of ATR leads to DNA damage and an increase of aberrant cells.** (A - B) Western analysis of whole cell extracts from mycATR and mycATRΔC-/- cells. Cells were left untreated (NT) or treated with 0.5 mM of Hydroxyurea (HU) for 20 hours (Chronic) or treated with 5 mM of Hydroxyurea (HU) for 5 hours (Acute), washed and resuspended in fresh media. The samples were collected at 0 and 5 hours after chronic

treatment, and 0 and 4 hours after acute treatment. Blots were probed with α-γH2A antiserum; α-EIF1α was used as a loading control (predicted protein sizes are indicated, kDa). (C – D) Quantification of signal from the western blot showed in (A and B) using ImageJ software; statistical analysis was conducted using Graph Prism (Error bars ± SD; n = 3 experiments, * p < 0.05; ** p < 0.005; *** p < 0.001; **** p < 0.0001. Unpaired t-test). (E) Representative images acquired by a LSM880 Zeiss confocal microscope from Z stack images of mycATRΔC-/- (cl8) cells after release from chronic treatment (5 hours post treatment). Cells were fixed with 3% formaldehyde, genomic DNA stained with DAPI (cyan) and nucleus (n), kinetoplast (k) and micronucleus (n*) are indicated; α-tubulin was used to stain the cytoskeleton (gray). Bar = 12 um. (F) Percentage of populations with the indicted proportions of Nucleus/Kinetoplast (n/k) in mycATR and mycATRΔC-/- cells untreated (NT) or after release from chronic (5 hours post treatment) or acute (4 hours post treatment) HU treatment. The populations were coloured according to the proportion of n/k as normal (1n1k, 1n2n, 2n2k) or aberrant (0n1k, 1n0k, 2n1k, the presence of a micronucleus, 1M and Others for low aberrant cells as showed 0K2N). Error bars ± SD, n = 2 (>120 cells counted/experiment).

Given an increase in the sub-G1 population after both chronic and acute HU treatment (Fig 3C), we characterised and quantified the types of aberrant cells presumed to arise based on the proportion of nuclear and kinetoplast staining in individual cells. Key events in the cell cycle of trypanosomatids include flagellum duplication, followed by the duplication and separation of the kinetoplast (mitochondrial DNA), and the duplication and separation of the nucleus before cell division [61,62]. To assess these events, we stained the DNA of fixed mycATR and mycATRΔC-/- cells with DAPI after HU treatment and analysed the proportions of nuclei (N) and kinetoplasts (K) in individual cells, whose outline was defined using α-Tubulin (Fig 4E). While only 1–3% of mycATR cells had aberrant N-K configurations after both chronic and acute HU treatment (Fig 4F), approximately 20% of mycATRΔC-/- cells exhibited aberrant N-K configurations after either HU treatment. The majority of these aberrant cells were classified as 0N1K, meaning they retained kinetoplast staining but lacked nuclear staining. These findings reveal that aberrant cell division, likely leading to the altered DNA content as seen by sub-G1 staining in FACS analysis (Fig 3C), arises after HU treatment in mycATRΔC-/- cells, and reiterates that ATR plays a role in *L. major* genome homeostasis after DNA replicative stress.

## ATR is needed to maintain the DNA replication program in *Leishmania major*

*L. major* exhibits a unique DNA replication program in which smaller chromosomes replicate before larger ones [63]. This pattern may be connected to observations that an early replication region (ERR) arises in each chromosome in proximity with a transcription switch strand region (SSR-Ori), which is consistently detected as a primary site of DNA replication initiation [64] as detected by MFA-seq [64,65]. Given that ATR is important to ensure the completion of DNA replication under both normal and replication stress conditions, we conducted MFA-seq analysis on both mycATR and mycATRΔC-/- cells, either untreated or treated with HU (Fig 5A) to assess if ATR plays a role during *Leishmania* DNA replication. Our work above revealed that exposing mycATR cells to chronic replication stress conditions permitted DNA replication to progress at a slow rate, while mycATRΔC-/- cells displayed a pronounced impairment in DNA replication in mid/late S phase with increased DNA damage and aberrant cells. For these reasons, we focussed on chronic replicative stress to test for replication timing defects.

To examine changes in DNA replication timing after ATR C-terminal deletion, we first plotted the MFA-seq profiles across individual chromosomes in the mycATRΔC-/- and mycATR cells (Figs 5B and S4), uncovering several key observations. First, the positive MFA-seq signal in mycATRΔC-/- cells appeared more broadly distributed across the larger chromosomes when compared with the MFA-seq signal from mycATR cells and in wild type *L. major*, where positive signal was predominantly seen around a single locus (Figs 5B and S4). Second, this broader distribution of MFA-seq signal locate outside the early replication region and was not entirely consistent between the two mycATRΔC-/- clones, suggesting stochastic variation in the changed DNA replication program (Figs 5B and S4). Finally, there was a less clear change in MFA-seq distribution on the smaller chromosomes relative to the larger, with the exception of chromosome 8, which showed a reduced MFA-seq signal around late regions of the mycATRΔC-/- clones compared with mycATR (S4 Fig).

Next, we assessed the correlation between DNA replication timing and chromosome size by linear regression analyses, using the average MFA-seq signal for each chromosome relative to its length (Figs 5C and S5). In the absence of

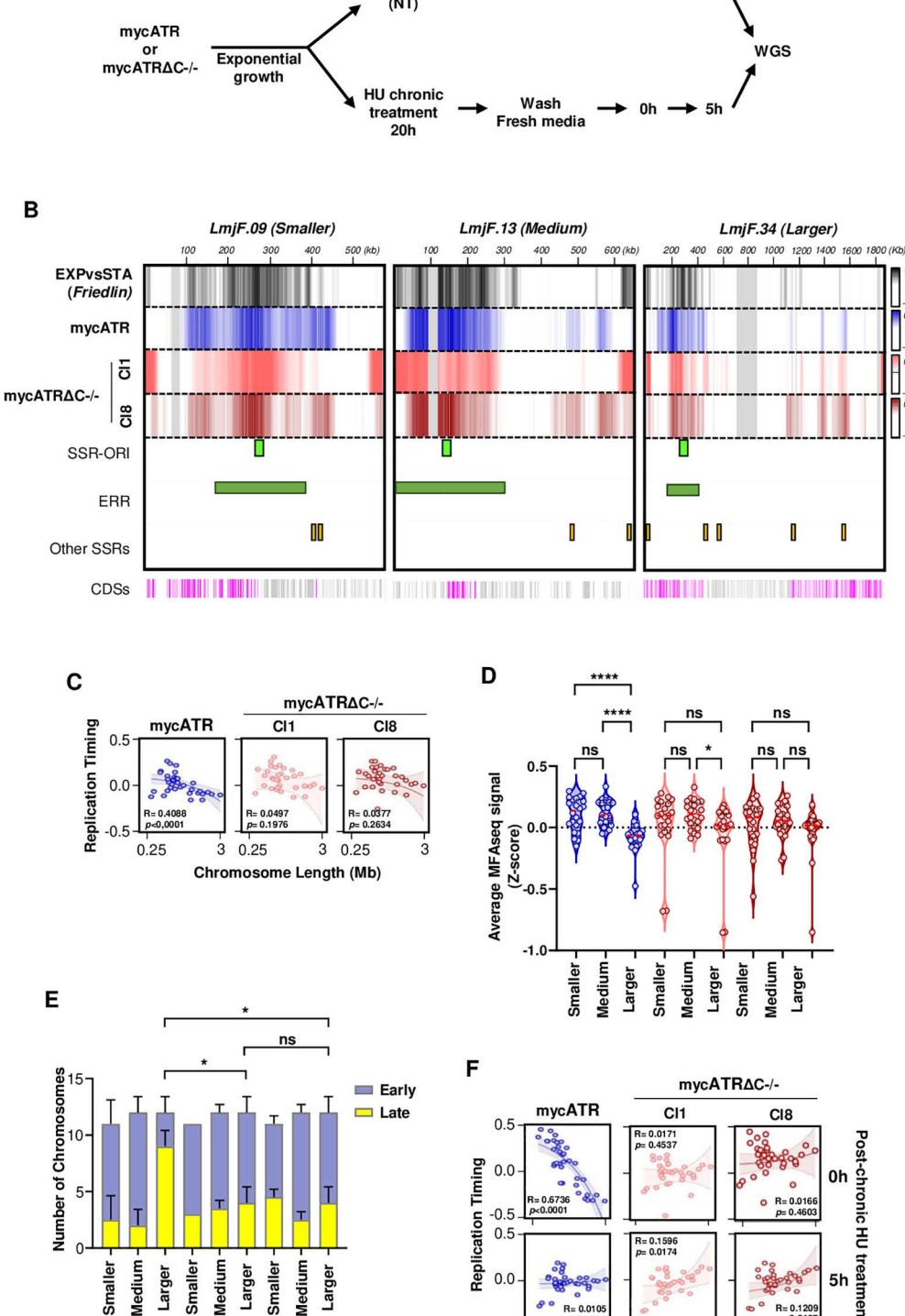

**Fig 5. ATR is a determinant of the *L. major* DNA replication programme.** (A) Schematic representation of the experimental design. mycATR or mycATRΔC-/- cells where left untreated (NT) or treated with 0.5 mM Hydroxyurea (HU) for 20 hours (Chronic). After treatment, cultures were washed and resuspended in fresh media. Time points were collected for whole genome sequencing (WGS) from NT cells and 0 and 5 hours post chronic treatment

release. DNA from a Cas9T7 cells at stationary (STA/ non replicative) phase cells were sequenced to allow MFA-seq analysis, which represents the ratio of read coverage in replicating samples relative to stationary cells (n = 2 experiments). This is figure was created using BioRender. (B) Representative snapshot of chromosomes 09 (Smaller), 13 (Medium) and 34 (Larger see classification in D), showing DNA replication timing (MFA-seq signal) using exponentially growing cells (untreated/NT), mycATR (blue) and mycATRΔC-/- (cl1 and cl8), light coral and firebrick, respectively; normalised with stationary cells (ratio); the signal represents the merge of the two replicates after normalisation (S4 Fig); positive (coloured) and negative (white) values indicate early and late replicating regions, respectively; the top heatmap (black) represents a previously published MFA-seq from wildtype L. major Friedlin [65]; SSR-Ori (light green), early replication regions (ERR) (dark green) and other SSRs (orange) in each chromosome are displayed; the bottom track indicates annotated CDSs (gray: transcribed from left to right; pink: transcribed from right to left), grey regions represent removed areas which have poor read mapping quality (see in Methods). (C) Linear regression analysis showing correlation between each chromosome size (x axis) and chromosome-averaged MFA-seq signal (DNA replication timing, y axis) of untreated mycATR and mycATRΔC-/- cells; R squared and P values are indicated at the left-bottom of each panel. (D) Quantification of the averaged MFA-seq in untreated mycATR and mycATRΔC-/- cells (values from both replicates). Chromosomes were sub-classified according with their length: Smaller (012345678910 – 11 and 14), Medium (12, 13, 151617181920212223 – 24) and Larger (252627282930313233435 – 36), Error bars ± SD; n = 2 experiments, p < 0.05; ** p < 0.005; *** p < 0.001; **** p < 0.0001, Unpaired t-test. (E) Quantification of the number of chromosomes that replicate according to MFA-seq average signal (replicate early, above 0; replicate late, below 0) in untreated mycATR and mycATRΔC-/- (cl1 and cl8) cells, Error bars ± SD; n = 2 experiments, p < 0.05; ** p < 0.005; *** p < 0.001; **** p < 0.0001. 2way ANOVA, and the significance checked with Turkey's HSD test. Chromosomes were sub-classified as in 5D. (F) Linear regression analysis showing the correlation between chromosome size (x axis) and chromosome-averaged MFA-seq signal (DNA replication timing/y axis) after chronic HU treatment (0 and 5 hours) in mycATR and mycATRΔC-/- cells; R squared and P values are indicated within each panel.

HU treatment, mycATR cells exhibited a chromosome size-DNA replication timing profile similar to that seen wild type L. major (Figs 5C and S5) [66]. Specifically, smaller and medium-sized chromosomes showed higher average MFA-seq signal (indicating earlier replication) compared with larger chromosomes, which exhibited lower average MFA-seq signal (indicating later replication). Remarkably, untreated mycATRΔC-/- cells showed no significant correlation between MFA-seq signal and chromosome size, suggesting a disrupted DNA replication program (Figs 5C and S5). This shift in replication timing was attributed to an increased MFA-seq signal (mean > 0) (Fig 5D) in several larger chromosomes (n = 7) in the mycATRΔC-/-cells, compared with only a few (n = 2) in mycATR cells (Fig 5E). Together, these findings indicate that loss of ATR's C-terminus results in earlier replication of larger chromosomes (Fig 5D and 5E). Interestingly, chromosomes 8, 12 and 31, exhibited significantly lower average MFA-seq signals compared with all other chromosomes, especially after replicative stress, in mycATRΔC-/- cells (S5 Fig). This observation suggests that these chromosomes may have distinct replication dynamics in these cells. Taken together, our data indicate that deletion of ATR's C-terminus disrupts the unique DNA replication timing of the L. major chromosomes, resulting in increased levels of DNA replication at several loci on the larger chromosomes. Interestingly, these changes in DNA replication programming do not appear to be associated with significant cell cycle defects (Fig 3C) or alterations in the proportion of replicating cells within the mutant population (S1B Fig).

To determine whether the changes in DNA replication timing observed in the ATR mutants were linked to DNA replication stress responses, we next analysed the replication timing profile of HU-treated mycATR and mycATRΔC-/- cells. We first performed linear regression analysis, again correlating the average MFA-seq signal of each chromosome with its length (Figs 5F and S5). At 0 hours, when mycATR cells display an increased proportion of the population in early S-phase (Fig 3C), the chromosome-size-dependent replication timing profile observed in untreated cells was maintained: smaller chromosomes had higher average MFA-seq signals compared with larger chromosomes (Figs 5F and S5). In contrast, after 5 hours, when most mycATR cells had traversed S-phase (Fig 3C), the MFA-seq signal was similar across both small and large chromosomes, indicating the completion of DNA replication (Figs 5F and S5). Loss of ATR's C-terminus again disrupted the DNA replication timing profile (Fig 5F). As observed in untreated cells (Fig 5F), mycATRΔC-/- cells at 0 hours showed no correlation between average MFA-seq signal and chromosome length. However, 5 hours after chronic HU treatment release, linear regression analysis revealed a reversal of the DNA replication program seen in unperturbed cells, with larger chromosomes showing higher average MFA-seq signals (Figs 5F and S5). These findings suggest that the combination of ATR loss and chronic replication stress leads to a severe disruption of the normal DNA replication program typically observed in unperturbed L. major promastigote cells, especially at later stages of S phase.

## ATR activity regulates DNA replication progression from early-S replicating loci and putative replication stress sites

To understand these alterations to DNA replication timing we describe above, we examined the MFA-seq signal across the genome under all conditions. The MFA-seq data revealed that, as previously reported, DNA replication in all chromosomes predominantly initiates at ERR, located at a strand switch region (SSR-Ori) where polycistronic transcription initiates and/or terminates, and proximal to the putative centromere [64,65,67,68]. To investigate DNA replication around this ERR, we generated metaplots of MFA-seq signal in mycATR and mycATRΔC-/- cells. In the absence of HU, we detected no quantifiable differences in MFA-seq signal between mycATR and mycATRΔC-/- cells at the 36 ERRs, whether analyzed collectively or separated by chromosome size (S6A and S6B Fig). This finding suggests that the altered DNA replication timing in the ATR mutants is not due to changes in replication initiation at these loci. We also generated metaplots of MFA-seq signal in mycATR and mycATRΔC-/- cells treated with HU. Unlike untreated cells, immediately after HU exposure, the metaplots showed a higher MFA-seq signal at the ERRs in mycATR cells compared to mycATRΔC-/- cells (Fig 6A and 6B). Furthermore, while the ERR signal decreased in mycATR cells after 5 hrs of HU treatment, it remained unaltered in mycATRΔC-/- cells (Fig 6A and 6B). These findings are consistent with the expected movement of DNA replication forks away from the ERRs from 0 to 5 hours (early S-phase to late S-phase; Fig 3C) in mycATR cells, and indicate that this progression is impaired in the absence of functional ATR in mycATRΔC-/- cells.

Since the observed impairment of DNA replication progression from the ERRs in the absence of functional ATR and under low levels of HU cannot explain the reversal of the DNA replication timing programme, we next examined MFAs-seq mapping across entire chromosomes (Figs 6C, S7 and S8). The impaired progression of DNA replication around the ERR in HU-treated mycATRΔC-/- cells compared with mycATR cells was readily apparent, as evidenced by a narrower positive MFAs-seq signal in the former. In addition, the profiles revealed the emergence of positive MFA-seq signal at several non-ERR sites in each chromosome after HU treatment. These new signals were more widespread after 5 hours of HU treatment than at 0 hours, and were more abundant in the mycATRΔC-/- cells than in mycATR cells (Figs 6C, S7 and S8). Visual inspection of the MFA-seq profiles revealed that, while some of these 'new' MFA-seq signals coincided with non-ERR SSRs, many did not. Thus, although predicted sites of RNA polymerase loading and unloading are known to be key locations of DNA replication stress in *L. major* [47], and given that collisions between the transcription and DNA replication machineries are a well-established trigger of ATR function [69], SSRs do not appear to be the sole locations of HU-induced MFA-seq signal accumulation.

Two types of repetitive sequences that are part of short, interspersed degenerate retrotransposons (SIDERs, which are involved in posttranscriptional (SIDER1) and translational (SIDER2) gene expression regulation) cover almost 5% of the *L. major* genome [40,70–72]. Similar to the SSRs, visual inspection suggested that some SIDER1 and SIDER2 sites in the genome correlated with HU-induced MFA-seq signal (Figs 6C, S7 and S8), but not all. To test if there is an overlap between SSRs and SIDER elements with the HU-induced MFA-seq signals, we first examined the distribution of SIDER1 and SIDER2 across the genome (Fig 6D and S1 Table). SIDER1 elements (n = 79) were less abundant than SIDER2 (n = 266) in non-ERRs. However, both types were more abundant on larger chromosomes. Notably, regions where either SIDER1 or SIDER2 was located within 5 Kb of an SSR were particularly common on the larger chromosomes (Figs 6D and S1). Given this distribution, we generated metaplots of the MFA-seq signal from mycATR and mycATRΔC-/- cells, both untreated or after HU chronic treatment, around all possible combinations of non-ERR SSRs and SIDERs (Fig 6E). These plots revealed that there was no detectable accumulation of MFA-seq signal at any of the three sequence features alone under the tested conditions, consistent with re-analysis of previous MFA-seq data in wild type *L. major* cells [65] (S9A Fig). However, a modest enrichment of the MFA-seq signal was observed only at sites where SSRs overlapped with either SIDER1 or SIDER2 in untreated mycATRΔC-/- and mycATR cells, and was most pronounced at divergent (*DIV*) and Head-to-Tail (*HT*) SSRs (Figs 6E, S9B and S9C). These HU-induced signals displayed a pronounced increase at 0 and 5 hours post-HU treatment and

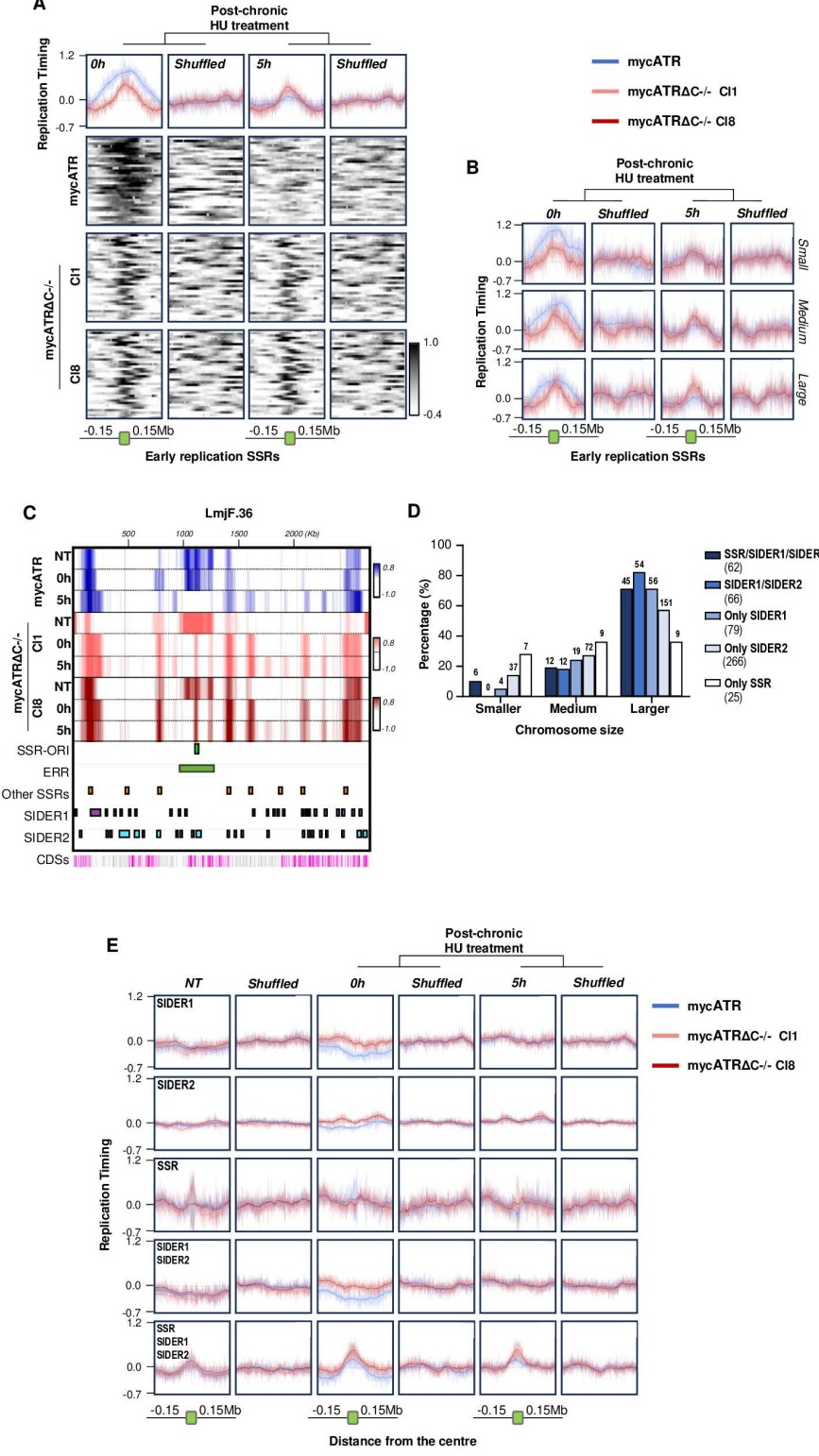

**Fig 6. ATR activity is important for DNA replication progression both from the early replicating locus in each chromosome and around putative replicative stress regions.** (A) Metaplots of global MFA-seq signal of mycATR (blue) and mycATRΔC-/- (cl1 and cl8) cells (light coral and firebrick, respectively) 0 and 5 hours post release from chronic HU treatment. The signal was plotted over the Early replication SSRs from each chromosome

(n = 36, Centre), including ± 0.15 Mb of upstream and downstream sequence; the line indicates the mean, and the light-coloured areas around the lines indicate the standard deviation between the two experimental replicas (SD); metaplots from shuffled genome regions were used as control. Below: colourmaps represent the MFA-seq signal in each early replicating SSR in each sample; metaplots and heatmaps from shuffled genome regions were used as control. (B) Metaplots of global MFA-seq signal, as in (A), after subclassifying the chromosomes according to length: Smaller (chromosomes 01 – 11 and 14), Medium (12, 13, 1516171819202122223 – 24) and Larger (25262728293031323334335 – 36); metaplots from shuffled genome regions were used as control. (C) Representative snapshot of chromosome 36 showing DNA replication timing (MFA-seq signal) using mycATR (blue) and mycATRΔC-/- (cl1 and cl8) cells (ight coral and firebrick, respectively) during exponentially growth (untreated/NT) or post-release from chronic HU treatment (0 and 5 hours); the signal represents the merge of the two replicates after normalisation (S8 Fig); positive (coloured) and negative (white) values indicate early and late replicating regions, respectively; SSR-Ori (light green), early replication regions (ERR) (dark green), other SSRs (orange), SIDER1 (purple) and SIDER2 (cyan) positions in the chromosome are displayed; the bottom track indicates annotated CDSs (gray: transcribed from left to right; pink: transcribed from right to left), grey regions represent areas with poor mapping quality. (D) Graphic showing the percentage and number of SSR/SIDER1/SIDER2, SIDER1/SIDER2, or any of the three features alone, in each category of chromosome size (smaller, medium and larger). (E) Metaplots of global MFA-seq signal, at 0 and 5 hours post-release of chronic treatment, in mycATR (blue) and mycATRΔC-/- (cl1 and cl8) cells (light coral and firebrick, respectively) plotted over the regions described in 6D (Centre, with ± 0.15 Mb of upstream and downstream sequence. Metaplots from shuffled genome regions were used as control.

in mycATRΔC-/- cells (Figs 6E, S9B and S9C). These findings suggest DNA replication at new loci after ATR loss and chronic HU exposure, particularly at loci on larger chromosomes, which may explain the disruption of chromosome size-dependent DNA replication timing.

## ATR is necessary for genome stability and variability in *Leishmania major*

Given that ATR partial deficiency exacerbates replication stress in *L. major* [47], we hypothesized that the loss of ATR function, along with the altered DNA replication program, might affect genome stability in *L. major*. To test this, we performed short-read Illumina whole genome sequencing of mycATRΔC-/- and mycATR cells after exposure to chronic HU treatment and subsequent culture for 10 passages in fresh media (approximately 30 generations) (Fig 7A). We compared cells in the initial (P0) and 10th passage (P10) to test for changes in single nucleotide Variants (SNVs), small insertions and deletions (InDels) and chromosome copy number changes (Copy Number Variants; CNVs), which can be associated with genome instability (S7B – S7D Fig). First, we measured the formation of SNVs and InDels, since both can result from replication errors or defects [73]. To do so, SNVs and InDels, in both mycATR and mycATRΔC-/- (cl1 and cl8) cells, in each condition (untreated and post HU chronic treatment release, P0 and P10), were measured relative to the reference genome using *FreeBayes* and *VCF-filter* (*see in Methods*). To isolate only the new markers that were formed during the period of growth, we selected only SNVs and InDels present in the P10 samples and not at P0, using *VCF-VCF filter* (*see Methods*, and scheme in Fig 7B); new counts were averaged in 1 Kb windows. The results showed that mycATR cells accumulated significantly more SNVs, in both conditions, than either mycATRΔC-/- clone (Fig 7B). In contrast, the accumulation of InDels was higher in the mycATRΔC-/- (cl1 and cl8) cells, especially after been submitted to a period of chronic HU treatment, where the accumulation was significantly higher when compared with mycATR cells under the same condition (Fig 7C). With these data we can correlate the accumulation of SNVs (which are potentially less harmful and more tolerated eukaryotes [74,75],including *Leishmania* [76–78] with the presence of an efficient DDR response (mycATR cells), while an increase of InDels (potentially more disruptive because they may cause frameshift mutations) [73,79,80] results from a DDR defect caused by the absence of a functional ATR, which might involve activation of alternative DNA repair pathways. However, further characterisation is needed to understand the mechanisms of DNA repair that generate such genome variations in *Leishmania*.

Next, we measured the levels of CNV in each cell and in each condition. To do that, we compared the $\log_2$ Ratio between the coverage of P10 samples over the coverage of P0 samples using *bamCompare* (*see Methods*, Fig 7C). The heatmap showed that chromosomes 8, 12 and 31 from at least one condition in the mycATRΔC-/- clones had a significantly decreased level of CNV when compared with the same chromosomes in mycATR cells (Fig 7C). Interestingly, those

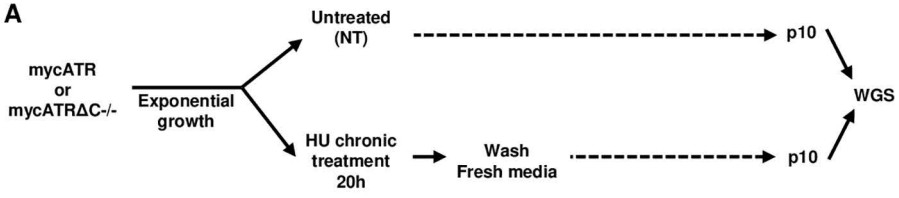

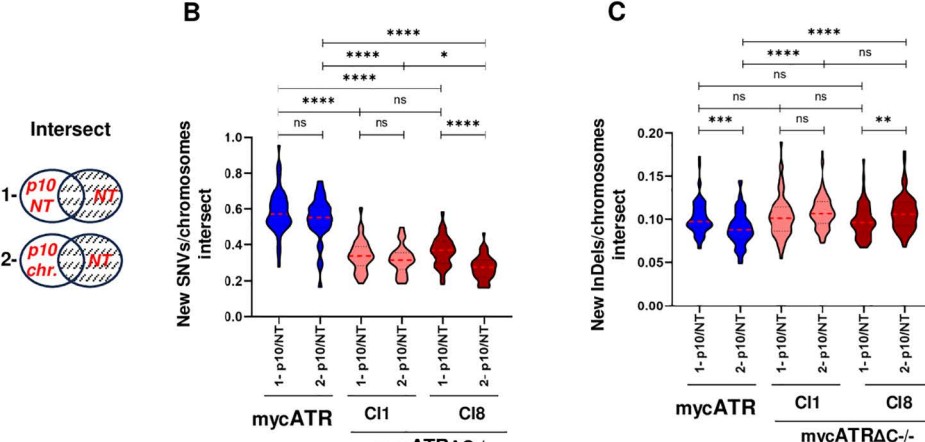

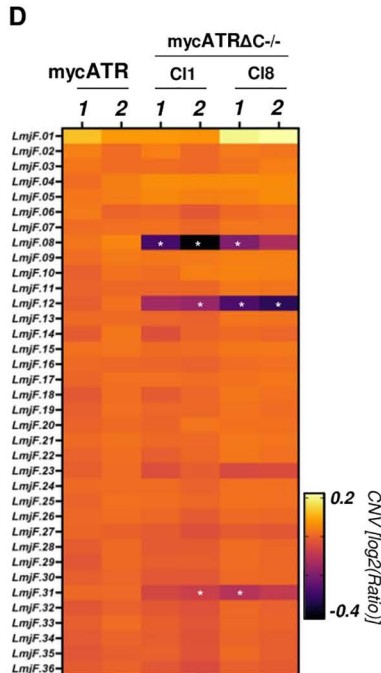

**Fig 7. ATR is necessary for genome stability and variability in *Leishmania major*.** (A) Schematic representation of the experimental design. mycATR or mycATRΔC-/- cells where left untreated (NT) or treated with 0.5 mM Hydroxyurea (HU) for 20 hours (Chronic). After treatment, cultures were washed and resuspend in fresh media for 10 consecutive passages before DNA extraction and WGS. This is figure was created using BioRender. (B)

Graphic showing normalized density of new SNVs and InDels (intersection scheme), respectively, after each condition in (A) for each cell. The scheme at the left side shows how the variations were calculated (only the unique SNVs or InDels, clear area, were considered). (C) Colourmap showing genome-wide relative copy number variation (CNV) analysis; chromosomes are ordered by size from top to bottom; CNV in mycATR and mycATRΔC-/- (cl1 and cl8) cells are expressed as $\log_2$[ratio(normalized reads from P10/normalized reads from P0)] for either 1- p10 NT or 2- p10 chronic conditions; chromosomes 08, 12, and 31 of mycATRΔC-/- (cl1 and cl8) cells are indicated with (*) meaning p < 0.05 in Unpaired t-test in comparison with mycATR respective chromosomes.

chromosomes are the same ones that showed a low MFA-seq signal (S5 Fig). Altogether, these data indicate that *L. major* ATR acts to promote genome stability (CNV) and variability (SNVs, InDels) after replicative stress. [47]

## Discussion

In this study, we provide a genetic dissection of ATR function in *L. major* genome maintenance, building on previous studies that utilized ATR chemical inhibition [44,45]. For reasons that remain unclear, we were unable to completely delete the *ATR* gene. Instead, we generated homozygous mutants lacking the C-terminus of ATR, which includes the predicted kinase domain. The truncation led to a significant loss of ATR function, manifested at least in part through reduced protein abundance and loss of nuclear localisation. The evidence of such loss of function rests on several observations of the mycATRΔC-/- cells we generated: impaired growth *in vitro*, aberrant cell cycle progression, accumulation of ssDNA, and increased levels of nuclear damage seen as increased yH2A levels. We propose that these phenotypes reflect ATR's conventional roles in managing DNA replication stress. This interpretation is consistent with the mutant's heightened sensitivity to HU and the exacerbation of many of the above defects after HU treatment. Notably, our data also reveals a role for ATR in controlling the unusual DNA replication program of *L. major*, wherein smaller chromosomes are duplicated earlier than larger chromosomes.

ATR plays a central role in the cellular response to DNA damage, utilizing its kinase activity to regulate numerous repair pathways [2,3,25]. One of the most well-studied pathways involves cell cycle control; for instance, ATR activates checkpoints in response to DNA damage, delaying progression through the S or G2 phases of the cell cycle [22,81]. Our findings suggest that ATR plays a similar role in *L. major* and attribute this function to the C-terminus of the protein. Deletion of this region impairs the correct distribution of nuclear and kinetoplast (mitochondrial) DNA during cell division and disrupts cell cycle progression in the presence of HU. Similar effects are observed when ATR expression is knocked down in *T. brucei* [42,43]; however, we cannot say here if these functions are solely dependent on kinase activity, as the truncated *L. major* ATR fails to efficiently localize to the nucleus. Nonetheless, this work contributes to the growing evidence that *L. major* ATR connects with wider factors needed to manage DNA replication progression. Previous studies showed that partial loss of Rad9 or Hus1 – components of the 9-1-1 complex – causes accumulation of cells in late S and G2 phases after acute HU treatment [82–85] and conditional deletion of Hus1 was shown to be lethal, resulting in the accumulation of ssDNA and γH2A – effects similar to those observed in mycATRΔC-/- cells [84]. However, further investigation is needed to clarify *Leishmania*'s response to replication stress and ATR's role in it. For instance, there are no reports of ATRIP or the ATR activators TOPBP1 or ETAA1 [86], and *L. major* 9-1-1 components appear to interact in distinct complexes during genome maintenance [87]. Still, our data confirm ATR's role in responding to replication stress across trypanosomatids.

An intriguing feature of *Leishmania* biology is its chromosome size-dependent DNA replication timing [66]. This work reveals that ATR is a determinant of *L. major* DNA replication timing, since its mutation leads to earlier replication of the normally late-replicating large chromosomes. The basis of this change in timing is revealed by MFA-seq, which shows that unlike in wild type cells, where DNA replication initiation is predominantly detected in early S-phase at just a single locus in each chromosome, MFA-seq peaks emerge at multiple sites across the chromosomes in the absence of functional ATR and, in particular, when the *L. major* ATR mutants are exposed to HU. It should be noted that MFA-seq is a population-level analysis and has relatively low resolution, since potential sites of initiation are inferred as the summit

of much wider peaks of sequence enrichment in replicating cells relative to non-replicating. For this reason, what we describe here may be only the most prominent new sites of replication initiation in ATR mutants or after HU treatment, and MFA-seq may overlook wider initiation loci, such as has been recently detected in wild type *L. major* by single-molecule DNAscent analysis (Damasceno et al BioRxiv 10.1101/2024.11.14.623610). Irrespective, one explanation for these new MFA-seq signals is that they represent normally suppressed origins, which are activated in the absence of ATR with and without excess replication stress caused by HU, and randomness in the loci at which they are activated could explain the differences in MFA-seq profiles observed in each chromosome in untreated conditions between the mutant clones (S4 Fig). Such a view would align with known functions of ATR in other eukaryotes, where licensed but not activated origins are controlled by ATR signalling [88], which acts as a negative regulator of origin firing and prevents inappropriate activation of origin firing in unperturbed cells [89], functions that are important to guarantee a sufficient supply of DNA precursors and replication factors for optimal fork progression [89]. Interestingly, such control seems to be particularly important during and after replicative stress in *Leishmania*, since the MFA-signal around all loci (overlap between SSR/SIDER1/SIDER2 and ERR) and the MFA-seq profiles in each chromosome was consistent among the mutant clones (Figs 6E and S8). So far, there has been little evidence for activation of 'dormant' origins in trypanosomatids in response to replication stress; in *T. brucei*, only one single putative site in one chromosome was found [90]. However, although all SSRs appear to bind one subunit of ORC, only ~25% are activated during unperturbed growth [91], and so SSRs may be a likely location for dormant origins that would be used in response to replication stress. However, no work to date has described the localisation of ORC in any *Leishmania* genome.

During DNA replication stress, ATR in other eukaryotes supresses origin firing globally, but allows dormant origins to be fired locally, preventing problematic replication across the genome [92]. This might be consistent with our observations, where MFA-seq signal increases in mycATR cells around the single ERR in each chromosome immediately after HU chronic treatment, with more limited MFA-seq signal increase across the chromosomes than is seen in mycATRΔC-/- cells (Fig 6A and 6B). However, our data do not reveal the SSRs in *L. major* to be ubiquitous sites of increased MFA-seq signal after HU exposure, as might be predicted from ORC localisation in *T. brucei*. Instead, the major MFA-seq signals outside ERRs we see emerging after ATR mutation and HU exposure are where an SSR and a SIDER element are in proximity. While it remains possible that these loci are dormant origins, as these may be untested sites of ORC recruitment, an alternative explanation is that these are genomic loci where DNA replication is especially prone to stalling, and this is exacerbated by the loss of ATR and exposure to HU. Hence, what we observe through MFA-seq may not be dormant origin activation, but sites of predominant DNA replication restart. This explanation would be consistent with two observations. First, the abundance of SIDERs in *Leishmania* has been associated with genome variation [30,71], consistent with them being problematic sequences for DNA replication. Indeed, it is possible that this is especially true around SSRs, as these are sites of RNA Polymerase loading and unloading, and hence SIDER proximity may increase clashes between transcription and DNA replication machineries. Second, the change in DNA replication timing we demonstrate here appears not to be limited to ATR function, since it is also seen after loss of RNase H1 or RAD51 [41,66], neither of which are known to directly coordinate origin activity: RNase H1 acts on RNA-DNA hybrids, which can form at sites of replication-transcription clashes, and RAD51 is the catalyst of DNA damage repair by homologous recombination, including at stalled replication forks [93–95]. Thus, the normal course of DNA replication programming in *L. major* appears to be influenced by a range of activities that influence the effective completion of genome duplication.

Clarification of which of the above scenarios explain the change in DNA replication timing in *L. major* ATR mutants could come from examining the roles of a wider range of the parasite's DNA repair factors in DNA replication timing. For instance, an increase of cells with aberrant DNA content is also observed after the conditional deletion of HUS1, suggesting intersection of 9-1-1 and ATR activities [84]. In fact, chromatin immunoprecipitation analysis of RPA1 and RAD9 showed that both ATR pathways factors accumulate around SSRs in unperturbed or during replicative stress cells [47].

It may also be valuable to examine the effects of ATM loss since this DNA damage kinase acts on DNA double strand breaks and directs recruitment of RAD51.

In this study, we reveal that deletion of ATR's C-terminus leads to lowered accumulation of SNVs, but a substantial increase of InDels after several rounds of cell multiplication, especially in conditions of replication stress, suggesting a link between DNA replication maintenance and genome variation. Whether this is due to the change we observe in the normal pattern of *L. major* DNA replication timing is currently unclear, but it has been demonstrated that loci where DNA replication initiates are prone to accumulate SNVs [96,97], and, indeed, the *L. major* ATR mutant struggles to progress/re-start replication after replicative stress. Furthermore, CNV analysis showed that chromosomes 08, 12 and 31 displayed the most substantial changes in content and were both notably distinct from the wider chromosomes in replication timing and showed low levels of MFA-seq signal increase after replicative stress. Though we do not know why these chromosomes differ, these data further suggest that deviation from the novel chromosome-size dependent pattern of DNA replication may have implications for stability. In conclusion, this work adds to the emerging picture of an intricate connection between DNA replication programming and genome variation in *Leishmania*.

## Methods

### Parasite culture and generation of ATR cell lines

Promastigotes derived from *Leishmania major* strain LT252 (MHOM/IR/1983/IR) were cultured at 26 °C in HOMEM medium supplemented with 10% heat-inactivated foetal bovine serum. Briefly, the primers were produced using the gene ID of *ATR*, LmjF.32.1460, on Leishgedit.net as described in [48] and the gRNA sequences for ATR C'terminal deletion were selected using GPP sgRNA Designer as described in [98]. For each fragment amplified (sgRNA and donor fragment) five PCR reactions were performed as follows using Phusion polymerase (NEB, #M0530S) (98°C/1min, 98°C/30s, 60°C/30s, 72°C/45s or 1:45min, 72°C/5min, 35 cycles). All PCR reactions were precipitated and pooled as follows: Add ETOH 100% (cold) 250ul; 3M Sodium acetate 10ul; Glycoblue 3ul; and store at-20 °C (minimum 30min). Afterwards, the DNA was centrifuged as before for 20 min, then washed in ETOH 70% (cold): 500 ul by centrifugation as before for 20 min. The pellet was air dried then resuspended in 20 ul of MilliQ water. For each transfection, 10 µl of the precipitated DNA was used. For recombination transfection or double tagging, 5x10$^5$ *L. major* Cas9 cells were resuspend in 100 µl of 1 × Tb-BSF buffer and pulsed using the X-100 program in the Amaxa Nucleofactor IIb (Lonza) [99]. After transfection, cells were resuspended in 10 mL of fresh media with 20% of FBS and recovered for at least 12 hours, then the appropriate selection drug was added, and the cells plated on a 96-well plate on the dilution of 1:6 – 1:72 – 1:864 for ~ 5 weeks. To generate $^{myc}$ATR and $^{3M}$ATR expressing cells, transformants were selected using 10 µg.mL$^{-1}$ puromycin (*Pur*) and 20 µg.mL$^{-1}$ hygromycin (*Hyg*). Homozygote $^{myc}$ATR cells were used in a second round of transfection to generate $^{myc}$ATR$^{\Delta C-/-}$ cells, which were selected using 10 µg.mL$^{-1}$ puromycin (*Pur*), 20 µg.mL$^{-1}$ hygromycin (*Hyg*) and 1 µg.mL$^{-1}$ neomycin (*Neo*). Integration into the expected locus was confirmed by PCR analysis. S2 and S3 Tables detail the cells and primers used in this study, respectively.

### Antibodies

Primary antibodies used in this study were α-myc clone 4A (Millipore, #05–724), α-EIF1α (Millipore, #05–235), α-BrdU clone B44 (BD Bioscience, #347580), α-tubulin Beta clone KMX-1 (Millipore, #MAB3408), and α-γH2A (1:1000) from rabbit serum as previously described [60]. α-Mouse IgG Alexa Fluor 488-conjugated (ThermoFisher, #A11001), goat α-Rabbit IgG IRDye 800CW-conjugated (Li-COR Biosciences) and goat α-Mouse IgG IRDye 680CW-conjugated (Li-COR Biosciences). was used as secondary antibodies.

## Western blotting

Whole cell extracts were prepared by collecting cells by centrifugation, washing with 1x PBS + 1x protein inhibitor (ThermoFisher, #S8830), resuspending in extraction buffer: 1xNuPAGE LDS Sample Buffer (ThermoFisher), 10% of 1x protein inhibitor (Thermo Fisher) and 5% β- mercaptoethanol; heating the samples to 95 °C for 7 minutes. Whole cell extracts were resolved on 4–12% gradient Bis-Tris Protein Gels (ThermoFisher) and transferred to Polyvinylidene difluoride (PVDF) membranes (GE Life Sciences) for 90 min, 25V. Membranes were first blocked with 5% non-fat dry milk dissolved in 1x PBS supplemented with 0.05% Tween-20 (PBS-T) for 1 h at room temperature. Next, membranes were probed with primary antibody: α-myc (1:5000), α-EIF1α (1:25000), α-yH2A (1:1000); overnight at cold room 4 °C, diluted in PBS-T supplemented with 2% non-fat dry milk. After washing with PBS-T, membranes were incubated with HRP-conjugated secondary antibodies in the same conditions as the primary antibodies. Finally, membranes were washed with PBS-T. To detect and visualize bands, membranes were incubated with ECL Prime Western Blotting Detection Reagent (GE Life Sciences) and exposed to detection in a SynGenne Pxi touch gel Imaging System.

## Immunofluorescence and detection of cells in S phase

$2x10^6$ exponential growing cells were collected by centrifugation, washed in 1.0 ml of 1xPBS and resuspend in 500 ul of 3.7% paraformaldehyde for 15 min at room temperature. After fixation, cells were washed in 1x PBS and 30 ul of the cell was adhered into poly-L-lysine coated slides. Next, cells were resuspended in 1xPBS supplemented with 0.7% TritonX-100 for 20 min. Primary antibody were used in 1x PBS + 3% BSA in following dilutions: α-myc (1:500), α-tubulin (1:1000) and α-BrdU (1:500); and 30 ul of antibody solution was used for 1 hour. After washing with 1x PBS + 3% BSA, Secondary antibodies were diluted as primaries in a dilution of 1:1000 for 45 min. DAPI with fluoromount-G (ThermoFisher, #00-4959-52) was used to stain the DNA. Images were acquired using a Zeiss Elyra Super resolution microscope, a Zeiss LSM880 confocal microscope or a Leica DMi8 microscope. Further image processing was performed with ImageJ software. For DNA synthesis detection with EdU; $2x10^6$ exponentially growing cells were left untreated or after chronic and acute hydroxyurea (HU) treatment incubated with 150 uM of EdU (ThermoFisher) for a pulse of 30 min. Slide preparation was as described above, but instead of primary antibody, EdU was detected using a Click-IT reaction (Invitrogen, #C10339).

## S phase detection and cell cycle profile using flow cytometry

Cells were incubated with 150 μM IdU (ThermoFisher) for 30 min (S phase detection) and then fixed at -20 °C with a mixture (7:3) of ethanol and 1x PBS for at least 16 hr. Next, cells were rinsed with washing buffer (1x PBS supplemented with 1% BSA) and the DNA denatured for 30 min with 2N HCl, followed by neutralisation with phosphate buffer (0.2 M Na2HPO4, 0.2 M KH2PO4, pH 7.4). Detection of incorporated IdU was performed with 1:1000 α-BrdU antibody (diluted in washing buffer supplemented with 0.2% Tween-20) for 1 hour at room temperature. After washing, cells were incubated with α-mouse secondary antibody conjugated with 1:10000 Alexa Fluor 488 (diluted in washing buffer supplemented with 0.2% Tween-20) for 1 hour at room temperature and then washed. Finally, cells were stained with 1xPBS supplemented with 10 mg.mL⁻¹ Propidium Iodide (PI) and 10 mg.mL⁻¹ RNAse A and filtered through a 35 mm nylon mesh. FACSCelesta (BD Biosciences) was used for data acquisition and FlowJo software for data analysis. Negative controls (omission of α-BrdU antibody during IdU detection step) were included in each sample and used to draw gates to discriminate positive and negative events, and at least 30000 cells were used to obtain the results.

## Replication timing profiling using Marker Frequency Analysis coupled with Illumina sequencing (MFA-seq)

Genomic DNA was extracted from exponentially growing cells, treated with chronic or acute HU treatment, and from stationary cells using DNeasy Blood & Tissue Kit (QIAGEN) and sequenced as 75 nucleotide paired-end reads. All

libraries were prepared using QIAseq FX DNA Library Kit (QIAGEN). Sequencing was performed at BGI (https://gtech.bgi.com/bgi/login) in a DNBSEQ-G400 sequencing platform. The Galaxy web platform (usegalaxy.org) [100] was used for most of the downstream data processing. For quality control and removal of adaptors, FastQC (http://www.bioinformatics.babraham.ac.uk/projects/fastqc/) and trimomatic [101] were used, respectively. Trimmed reads were mapped to the reference genome (*Leishmania major* Friedlin v39, available at Tritrypdb - http://tritrypdb.org/tritrypdb/) using BWA-mem [102]. BamCompare (DeepTools) was used to determine read abundance from exponentially growing cells or after HU treatment relative to the reads from stationary culture. Ratios (MFA-seq signal) were first calculated in 500 bp consecutive windows using the reads counts method for normalisation; 169 areas with poor mapping were excluded [67]. Raw ratios were further transformed into Z scores relative to the whole genome ratio average as calculated in 100 kb sliding windows. S4 Table lists all the samples used in both replicas.

### CNV average

To calculate CNV ratios, BAM files of each sequence depth was calculated in 2 kb bins using bamCompare following the equation $\log_2$[ratio(normalized reads from P10/normalized reads from P0)]. The CNV average of each chromosome as calculated for each condition.

### SNV density

SNVs and InDels relative to the reference genome were detected in *p10* of each cell line in each condition and *p0* (untreated) cells using *FreeBayes* [103]. Only those SNVs and InDels in regions with read depth of at least 5, with at least 2 supporting reads, and a map quality of 30 were considered. To better capture the genomic variability in the time frame of the experiments, variants present simultaneously in *p10* and *p0* cells were excluded from the analysis using VCF-VFC intersect function from VCFtools package [104].

### Graphs plots generation and statistical analysis

MFAseq signal snapshots were obtained from Integrative Genomes Viewer (IGV). Heatmaps and metaplots were generated with deepTools plotHeatmap and plotProfile tools, respectively, on Galaxy. Shuffled regions were generated using *bedtools shuffled* on Galaxy, where the shuffled regions have the same distance range as the areas of interest. Graphs were generated and statistical analyses were performed using Prism software. 2way ANOVA, and the significance checked with Turkey's HSD test was used when comparing more than two groups. Comparisons between the two groups were made with the student's Unpaired t-test. Figure legends include P-values, sample size, and experiments were repeated at least 2 times using biologically independent.

### Supporting information

**S1 Fig. Characterisation of ATR kinase mutants in *Leishmania major*.** (A) Growth curve of the indicated cells cultivated in HOMEM medium; cells were seeded at ~2x10^5 cells.ml^-1 at day 0; growth was evaluated for 6 days and cell density assessed every 24 hours; CC1 and Cas9T7 cells were used as control (Error bars ± SD; n = 2 experiments). (B) Representative pseudo-colour plots from a flow cytometry analysis to detect DNA synthesis in mycATR and mycATRΔC-/- (cl1 and cl8) cells. Cells were seeded at ~2x10^5 cells.ml^-1 at day 0; at the indicated time points an aliquot of each cell line was incubated with IdU for 30 min and IdU detected using α-BrdU under denaturing conditions; DNA was stained with propidium iodate; ~30,000 cells were analysed per sample; 1N and 2N indicate single and double DNA content; dashed red lines indicate the threshold used to discriminate negative from IdU-positive events; inset numbers indicate total percentage of IdU-positive events in the whole population (n = 2 experiment).
(PDF)

**S2 Fig. The ATR C-terminus is vital for accumulation of the kinase in the nucleus.** (A) Schematic representation of the region where we predict a possible Nuclear Location Signal (NLS). Predicted ATR protein sequence was accessed through Tritryp.pdb (LmjF.32.1460), and possible NLS signals assessed using two available software (http://nls-mapper.iab.keio.ac.jp/cgi-bin/NLS_Mapper_y.cgi) and (https://nucpred.bioinfo.se/cgi-bin/single.cgi). (B) Representative images of the sub-cellular localization of mNeongreen signal, which is N-terminally added to ATR in mycATR cells; scale = 13 µm. Images were captured using a SP6 microscope (Leica). DNA, shown in cyan, was stained with PBS and Hoescht solution for 10 min at 27°C followed by image acquisition; the mNeongreen signal is in yellow. (C) Representative images acquired on a LSM880 Zeiss confocal microscope from Z stack images of 3MATR (cell line with only 3x myc tag at N'terminal of ATR gene). Cells were fixed with 3% formaldehyde and stained with α-myc in yellow; genomic DNA stained was with DAPI in gray (Bar = 11 um).
(PDF)

**S3 Fig. Replication stress leads to low DNA synthesis intensity in mycATRΔC-/- cells.** EdU intensity (A.U.) from positive EdU mycATR or mycATRΔC-/- cells in each condition showed in Fig 3E. An area was drawn around each EdU positive cell's nucleus based on the DAPI stain and the EdU signal intensity was measured using ImageJ software, (Error bars ± SD; n = 2 experiments, p < 0.05; ** p < 0.005; *** p < 0.001; **** p < 0.0001. Unpaired t-test).
(PDF)

**S4 Fig. ATR is essential to maintain the replication program in *Leishmania major*.** Representative snapshots of all chromosomes (excluding chromosome 01, due an amplification in the STA control), showing DNA replication timing (MFA-seq signal) using exponentially growing (untreated/NT) mycATR (blue) and mycATRΔC-/- (cl1 and cl8) cells (light coral and firebrick, respectively). Signals in two replicates (rp1 and rp2) and the merged signal for each cell are shown; positive (coloured) and negative (white) values indicate early and late replicating regions, respectively; the top heatmap (black) represents a previous published MFA-seq from wildtype *L. major* Friedlin [65]; SSR-Ori (light green), early replication regions (ERR) (dark green) and other SSRs (orange) positions in each chromosome are displayed; the bottom track indicates annotated CDSs (gray: transcribed from left to right; pink: transcribed from right to left), grey regions represent removed areas which have poor mapping quality (*see Methods*).
(PDF)

**S5 Fig. ATR is essential to maintain the replication program in *Leishmania major*.** Linear regression analysis for each replicate (rp1 and rp2), showing correlation between chromosome size (x axis) and chromosome-averaged MFA-seq signal (DNA replication timing, y axis) in untreated (NT) mycATR and mycATRΔC-/- (cl1 and cl8) cells post-release from chronic HU treatment (0 and 5 hours) in; R squared and P values are indicated within each panel.
(PDF)

**S6 Fig. ATR activity is important for DNA replication progression both from the early replicating locus in each chromosome and around putative replicative stress regions.** (A) Metaplots of global MFA-seq signal in mycATR (blue) and mycATRΔC-/- (cl1 and cl8; light coral and firebrick, respectively) cells, 0 and 5 hours post release from chronic HU treatment; the black line represents previously published MFA-seq from wildtype *L. major* Friedlin [65]. Signal was plotted over the Early replication SSRs from each chromosome (n = 36, Centre) ± 0.15 Mb of upstream and downstream sequence; the line indicates the mean, and the light-coloured areas around the line indicates the standard deviation between the two experimental replicas (SD); metaplots from shuffled genome regions were used as control. Below: the colourmaps represent the MFA-seq signal in each early replicating SSR in each sample; metaplots and heatmaps from shuffled genome regions were used as control. (B) Metaplots of global MFA-seq signal, as in (A), after subclassifying the chromosomes according to length: Smaller (chromosomes 01 – 11 and 14), Medium (12, 13,

151617181920212223 − 24) and Larger (252627282930313233435 − 36); metaplots from shuffled genome regions were used as control.
(PDF)

**S7 Fig. ATR activity is important for DNA replication progression both from the early replicating locus in each chromosome and around putative replicative stress regions.** Representative snapshots of all chromosomes (excluding chromosome 01 due an amplification on the STA control), showing DNA replication timing (MFA-seq) in mycATR (blue) and mycATRΔC-/- (cl1 and cl8light coral and firebrick, respectively) cells that were either exponentially growing (untreated/NT) or post-release from chronic HU treatment (0 and 5 hours); the signal represents the merge of the two replicates after normalisation (S8 Fig); positive (coloured) and negative (white) values indicate early and late replicating regions, respectively; SSR-Ori (light green), early replication regions (ERR) (dark green), other SSRs (orange), SIDER1 (purple) and SIDER2 (cyan) positions are displayed; the bottom track indicates annotated CDSs (gray: transcribed from left to right; pink: transcribed from right to left), grey regions represent removed areas which have poor mapping quality.
(PDF)

**S8 Fig. ATR activity is important for DNA replication progression both from the early replicating locus in each chromosome and around putative replicative stress regions.** Representative snapshots from the experimental replicas used in S7 Fig (DNA replication timing (MFA-seq) in mycATR (blue) and mycATRΔC-/- (cl1 and cl8, light coral and firebrick, respectively) cells that were either exponentially growing (untreated/NT) or post-release from chronic HU treatment (0 and 5 hours)).
(PDF)

**S9 Fig. ATR activity is important for DNA replication progression both from the early replicating locus in each chromosome and around putative replicative stress regions.** (A) MFA-seq signal from a previous report [65] in wildtype cells (*L. major* Friedlin), plotted around the regions described in (6D) ± 0.15 Mb of upstream and downstream sequence. (B) Graphic showing the percentage and number of SSR/SIDER1/SIDER2, or SSRs alone (Only SSR) in each type of Strand Switch Region (convergent, divergent and head-to-tail). (C) Metaplots of global MFA-seq signal in mycATR and mycATRΔC-/- (cl1 and cl8) cells after chronic HU treatment (0 and 5 hours) around each type of SSR (*COnVergent, DIVergent, HT head-to-tail*) that are in proximity with either SIDER1 or SIDER2 ± 0.15 Mb of upstream and downstream sequence. Metaplots from shuffled genome regions were used as control.
(PDF)

**S1 Table. Table detailing all 36 ERRs described in [64 and 65] showing the start, end and length of the region (Green); regions described in Fig 6D where we observed the overlap between SSR and the two types of repetitive sequences, SIDER1 and 2, used to generate the Metaplots in Figs 6E and S9C, showing the start, end and length (Orange), the highlighting regions where the signal is stronger; regions described in Fig 6D where we observed overlap between the two types of repetitive sequences, SIDER1 and 2, used to generate the Metaplots in Figs 6E and S9C, showing the start, end and length (Red).**
(PDF)

**S2 Table. Table showing all the cells generated for this study.**
(PDF)

**S3 Table. Table showing all primers used in this study.**
(PDF)

**S4 Table. Table showing the ID and the description of the samples sequenced by WGS.**
(PDF)

**S1 File. Numeric data underlying main and supplementary figures, which are identified by tabs.**
(PDF)

**S2 File. Western blot and flow cytometry data underlying Figs 1, 3 and 4.**
(PDF)

**S3 File. Text file of script for sequence analysis.**
(TXT)

## Acknowledgments

We thank the Molecular and Cell Biology of FMRP/USP staff and the University of Glasgow Centre for Parasitology for the support during the development of this project.

## Author contributions

**Conceptualization:** Gabriel L. A. da Silva, Jeziel D. Damasceno, Richard McCulloch, Luiz R. O. Tosi.

**Data curation:** Gabriel L. A. da Silva.

**Formal analysis:** Gabriel L. A. da Silva, Jeziel D. Damasceno, Jennifer A. Black, Richard McCulloch, Luiz R. O. Tosi.

**Funding acquisition:** Richard McCulloch, Luiz R. O. Tosi.

**Investigation:** Gabriel L. A. da Silva, Craig Lapsley.

**Methodology:** Jeziel D. Damasceno, Jennifer A. Black, Craig Lapsley, Richard McCulloch.

**Project administration:** Richard McCulloch, Luiz R. O. Tosi.

**Software:** Jeziel D. Damasceno.

**Supervision:** Jeziel D. Damasceno, Richard McCulloch, Luiz R. O. Tosi.

**Visualization:** Gabriel L. A. da Silva.

**Writing – original draft:** Gabriel L. A. da Silva, Richard McCulloch.

**Writing – review & editing:** Gabriel L. A. da Silva, Jeziel D. Damasceno, Jennifer A. Black, Richard McCulloch, Luiz R. O. Tosi.

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
