## [Decision Letter · Decision Letter 0]

23 Feb 2025

PGENETICS-D-25-00053

ATR, a DNA damage kinase, modulates DNA replication timing in Leishmania major.

PLOS Genetics

Dear Dr. McCulloch,

Thank you for submitting your manuscript to PLOS Genetics. After careful consideration, we feel that it has merit but does not fully meet PLOS Genetics's publication criteria as it currently stands. Therefore, we invite you to submit a revised version of the manuscript that addresses the points raised during the review process.

Please submit your revised manuscript within 60 days Apr 24 2025 11:59PM. If you will need more time than this to complete your revisions, please reply to this message or contact the journal office at plosgenetics@plos.org. Please include the following items when submitting your revised manuscript:

We look forward to receiving your revised manuscript.

Kind regards,

Dmitry A. Gordenin, Ph.D.

Academic Editor

PLOS Genetics

Geraldine Butler

Section Editor

PLOS Genetics

Aimée Dudley

Editor-in-Chief

PLOS Genetics

Anne Goriely

Editor-in-Chief

PLOS Genetics

**Journal Requirements:**

https://journals.plos.org/plosgenetics/s/submission-guidelines#loc-parts-of-a-submission

- TM on pages: 20, and 21.

5) We have noticed that you have uploaded Supporting Information files, but you have not included a list of legends. Please add a full list of legends for your Supporting Information files after the references list.

2) State what role the funders took in the study. If the funders had no role in your study, please state: "The funders had no role in study design, data collection and analysis, decision to publish, or preparation of the manuscript.".

7) Please ensure that the funders and grant numbers match between the Financial Disclosure field and the Funding Information tab in your submission form. Please note that the funders must be provided in the same order in both places as well. Currently, the order of the grants is different in both places.

**Comments to the Authors:**

**Please not that one of the reviews is uploaded as an attachment.**

**Reviewers' comments:**

Reviewer's Responses to Questions

Reviewer #1: This manuscript characterises the ATR checkpoint kinase homologue in the kinetoplastid parasite Leishmania. DNA replication, DNA damage checkpoints and DNA repair in this species is arguably a topic of particular interest because Leishmania has a highly plastic genome, and may have an unusual programme of genome replication timing.

The checkpoint kinases have been examined before in Leishmania using chemical inhibitors (e.g. PMID: 35059327, PMID: 30265735) and a monoallelic rather than diallelic ATR deletion (ref 47 preprint); plus the replication timing programme has been examined before via MFAseq by these authors. Overall, this somewhat limits the novelty of this manuscript: a large number of papers from the same authors, piecing together slightly different approaches to ATR, could arguably have been published in a more streamlined manner. Nevertheless, some advances are made here. The innovations are essentially a) to make a partial knockout of the ATR gene (deleting the kinase domain of both alleles) rather than chemically inhibiting it or knocking out only 1 allele, and b) combining this knockout with replication experiments via MFAseq.

The data are thoroughly analysed (modulo comments below) and the conclusions are largely justified by the data. They show that in this early diverging parasite, ATR appears to act broadly as it does in other eukaryotes, protecting stressed replication forks and promoting their recovery, despite the lack of obvious homologues for some ATR partners. The weakness of the paper is primarily the inability to define exactly how ATR acts – it is promotion of inefficient origin firing, or promotion of replication restart, or both? This is due to the low resolution of MFAseq, which allows only broad, indirect conclusions about DNA replication. Since higher-resolution techniques are available, e.g. via DNAscent technology (already used by this group in Leishmania) and DNA combing (used by other groups in the same parasite), it is not clear why MFAseq with its acknowledged weaknesses was chosen. A more succinct and direct manuscript could perhaps have been written if the data were more directly informative. An origin map via ORC ChIP could also greatly enhance the clarity about what is going on in Leishmania DNA replication. These are probably not, however, practical experiments to do or re-do at the revision stage.

Specific comments:

At L222 ‘Quantification of the signal intensity confirmed a significant increase in the HU-treated mycATR.C-/- cells (Figure 3B), indicating that loss of ATR leads to the accumulation of ssDNA in L. major’. Strictly, this appears to be overinterpreted: there are other explanations for extra BrdU staining that aren’t addressed. For example, the mutant parasites could have replicated more slowly, from different oris, or just overall more, during the IdU pulse – leading to more baseline IdU, although the same proportion of ssDNA.

L330 ‘Second, this broader distribution of MFA-seq signal was not entirely consistent between the two mycATR.C-/- clones, suggesting stochastic variation in the changed DNA replication program.’ Looking at Fig S4, the two clones are actually radically different, with cl8 resembling parental on many chromosomes, whereas cl1 does not. In general, the marked differences between the two clones – which are seen in several figures, not just here, e.g. in differing baseline ssDNA levels in Fig 3B – are neglected in the discussion. Why are 2 supposedly identical clones so different in phenotype?

L365: ‘the chromosome-size-dependent replication timing profile observed in untreated cells was maintained: smaller chromosomes had higher average MFA-seq signals compared with larger chromosomes…’ In general, I think the authors might acknowledge the fact that the somewhat mysterious ‘later replication timing’ of larger chromosomes could simply be due to them having a greater number of stochastically firing origins, as presumably required to get larger chromosomes replicated within a single S-phase. Pre-printed data from DNAscent (same authors) does, I believe, suggest this, and provides a much more conventional explanation for how the small and large chromosomes replicate in these parasites. After HU, larger chromosomes having higher MFAseq could also be consistent with stress simply leading to more stochastic ori firing, in an attempt to rescue replication? Overall, in the discussion (L484) this is presented as ‘our data also reveals a novel role for ATR in controlling the unusual DNA replication program of L. major’ – but I’m not sure this isn’t really just a side-effect of a ‘normal’ role for ATR.

In the final section re. genome mutations in ATR mutants, it seems contracdictory that ATR mutants get fewer SNPs, indels etc. but are supposed to fire more oris, since it also stated ‘loci where DNA replication initiates are prone to accumulate SNPs’??

L524 The discussion states: ‘So far, there has been little evidence for ‘dormant’ origins in trypanosomatids’… this appears to neglect both the DNAscent and DNA combing data, which showed plenty of stochastically firing origins?

In Fig 1, it would be useful to show how the kinase domain was defined.

Fig 4E: the microscopy pictures are very small and resolution appears poor – can the nuclei and kinetoplastids really be confidently defined?

Fig 5: did the authors really wait 10 cycles after HU stress before doing the MFAseq? If so, why? Surely the next cycle would be the place to see acute effects upon replication timing?? Or did they only wait 10 cycles before sequencing for mutations? This is not clear.

In Fig 5 there is a huge unexplained grey region with no apparent data in chr 8??

Fig 5D and 5E appear essentially redundant?

Fig S4 legend refers to ?KKT1 but doesn’t define it, and SIDERs are mentioned but not apparently labelled

Fig S7 : Chr 1 and 36 are absent, and there is a (D) in the legend after (A) but no fig (D) is seen?

Fig S8: is the data on CNVs at all quantifiable? Between the CNV changes both up and down, it does not look statistically significant.

Formatting comments:

L39 add comma after family

L74 add comma after ultimately

L80 accumulation?

L385 chromosomes (plural)

At L407, Fig S7 appears before he latter half of Fig S6, which is confusing

L409 correct ‘MFAs-seq’

L421: ‘SIDER’ appers but the abbreviation is not apparently defined?

L430: kb not Kb

L542: Fig 5A appears to be misreferenced (doesn’t show this)

L529: suppresses

Throughout the methods, check for failed italicisation of scientific names, use of u instead of mu for micro, lack of superscripts and subscripts, erratic capital letters, etc.

L1040 survival

L1076 ‘…I n each condition was calculated by the number of stained cells by the number of

present in each sample’ – sentence makes no sense?

L1105: ‘exponentially untreated growing cells’ – makes no sense?

L1116: delete in

L1117: ‘Chromosomes were sub-classified according with their length’ – what were the cutoffs?

L1125: ‘rp1 and rp2 represent the two replicas’ – these are nowhere on Fig 5

Reviewer #2: This paper studies the role of ATR in the trypanosomatid Leishmania major, which harbors an unusual chromosomal organization with early-replicating small chromosomes and late-replicating large chromosomes. The main contribution is the generation and characterization of a CRISPR deletion of the ATR kinase domain. The authors show that loss of the kinase domain leads to slow growth, HU sensitivity, ssDNA accumulation and aberrant cell cycle progression and DNA damage accumulation under replication stress. The most novel result is that loss of ATR activity leads to advancement of replication of most large chromosomes, and even more profoundly, under replication stress the large chromosomes replicate earlier than the small ones. This advanced replication appears to be driven by new initiation sites. Aside from several concerns detailed below, I found this to be an interesting and well described manuscript.

Specific comments:

- Figure 3C: the sub-G1 population is unusual as it shows a distinct peak and in some cases, more than one peak. This potentially suggests the loss of specific parts of the genome. Can the authors further characterize these patterns?

- Difference in MFA signal between the two clones- it would be useful to determine if this is a clone-specific difference or stochastic variation unrelated to the clones, in other words, if the same clone was profiled twice, would it also show the same variation? The legend of figure 5 suggests that two repetitions were done for each clone- can the consistency between them be quantified? Some of the differences between the clones seem concerning, for instance on the right side of chromosome 08, the two clones almost appear to replicate in opposite ways. The presentation using a heatmap is also somewhat confusing as it is difficult to evaluate differences along the early-late continuum and also difficult to compare to the Friedlin data that is shown as a graph rather than a heatmap. What is the correlation of the MFA data between experimental repetitions, between clones, and to the Friedlin data? Overall, the centrality of these results warrant a clearer presentation and a more rigorous analysis of data quality.

Minor: Figure 5C (and others) indicate a positive R, however the regression slope is negative. Does R need to be negative? Or is this R squared?

- The CNV and SNP analysis appears less rigorous, less well described, and less well controlled and validated than other analyses. Overall, I find this section not very convincing. What was the allele frequency used to call SNPs (as an aside, the term SNPs is incorrect and should be replaced with SNV or mutations)? Is 10 generations sufficient to accumulate mutations in the absence of selection? Have the cells been cloned in any way? The called SNPs (and CNVs) could well be subclonal or even sequencing mistakes. Have any of them been validated (e.g., Sanger sequencing)? Overall, this section should be more rigorous, or removed.

Reviewer #3: The authors made a study of the ATR protein of L. major. They failed to make a complete knockout of the ATR gene, but they tagged this gene at the N-terminus. However, the authors generated ATR mutants that lacked the predicted kinase domain at the C-terminus, and they analyzed these mutants and challenged them with hydroxyurea treatments. They found that the deletion of the C-terminus reduced the presence of the muted ATR protein in the nuclei. Mutants with deleted C-terminus showed aberrant cell cycle progression, accumulation of ssDNA in the nuclei and increased levels of yH2A, indicating increased levels of DNA damage. MFA-seq analysis was performed and the results showed that after ATR C-terminal deletion there are changes in the timing of DNA replication initiation. The authors also showed that deletion of the ATR C-terminus leads to a reduced accumulation of mutations, suggesting that ATR promotes chromosomal instability. The text is generally well written and understandable, but there are some statements that could be improved.

This study is useful for understanding the complex biology of this unicellular eukaryote of medical and veterinary interest. This work is also useful for understanding the processes of DNA replication stress and DNA damage response in this parasite. However, there are several deficiencies in the study that must be addressed to make it suitable for publication.

Major

- The statement that 'only a single locus per chromosome is consistently detected as the primary site of DNA replication initiation as detected by MFA-seq and DNAscent' is often repeated in the text. However, the MFA-seq methodology used in this study does not have the resolution of origin mapping to make this claim. It may be that there are many sites of DNA replication initiation within the large region that the authors refer to as the primary site of DNA replication initiation. It would be beneficial to consider moderating the interpretation of MFA-seq results to ensure a balanced and nuanced understanding of the findings.

Even more, it would be nice to discuss how and why the origins detected by MFA-seq in tagged mycATR differ from those detected in the WT strain that were previously published, saying that there is only one origin per chromosome in Leishmania (PMCID: PMC4612428). It seems that MFA-seq now detects more than one origin per chromosome even in the L. major WT strain (Figure S4).

- It would be very useful to include a supplementary table with the exact genomic locations of the detected MFA peaks in WT, mycATR and mycATRΔC-/- mutants. The chromosome sizes and lengths of the MFA peaks should also be included in this table. Any statements in the main text about relationships between broader or narrower MFA peaks and chromosome lengths should also be correlated with this supplementary table. The exact positions of the 36 ERRs should also be provided. Otherwise, this section of the text may prove challenging to comprehend.

- The authors should further discuss the choice of analyzing DNA replication in chronic HU. According to Figure 3C-E, DNA replication is perturbed but recovered after acute conditions, whereas DNA replication seems to be blocked after acute HU treatment (Figure 3C) and the observed minority of cells labelled with EdU could only be cells where DNA repair occurs (less than 5% of cells labelled with EdU in chronic HU condition according to Figure 3E). Furthermore, as shown in Figure 1E, 0.5 mM HU is lethal even for the mycATR strain after 4 days. This concentration is highly toxic and 5 mM used in acute condition is even more toxic. It would be useful if the authors could comment on this.

- The catalytic domain of LmjF.32.1460 is located at 2863-3135 aa according to the NCBI database. The authors deleted 2060-3135 aa and should comment on this.

- The authors state in the text (line 320-322): ‘Our work above revealed that exposing mycATRΔC-/- cells to chronic replication stress conditions permits DNA replication to progress, while mycATRΔC-/- cells displayed a pronounced impairment in DNA replication after acute HU treatment (Figure 3C)’. According to Figure 3C (and Figure 3E), the marked impairment of DNA replication (S phase block) is more pronounced in the chronic condition. It would be good to change the text accordingly.

- Authors should include shuffled controls (random genomic regions of equal size) for all metaplots (Fig 6A, B, E and S6).

- If it is feasible, authors should provide a statistical test for copy number variations (Figure S8) to support the claim that there is a modest increase in CNV levels on several chromosomes in both conditions (line 454 -455).

- Supplementary figure legend appears to contain some errors. It would be beneficial to consider re-checking and correcting these.

Minor

- One-way ANOWA statistically tests three or more groups of measurements. It would be useful if the authors could indicate which test was used to compare two measurements, as in Figure 3E, Figure 4C-D etc.

- The statistical test between untreated and HU-treated mycATRΔC-/- mutants should also be shown in Figure 3B.

- The authors should explain what exactly is shown on the y-axis (replication time) in Figure 5C. Is it the ratio of the average MFA-seq signal for each chromosome to its length? If so, they should include this explanation in the legend of Figure 5C. They should do the same for the average MFA-seq signal and how they quantify MFA signals.

- In the text the authors state (line 220-222) ‘Microscopy images showed a clear increase of nuclear IdU signal in HU-treated mycATRΔC-/- cells compared with untreated control, and with both treated and untreated mycATR cells (Figure 3A)’. However, mycATR cells are not shown in Figure 3A.

- Line 351-353 - chromosomes 8 and 31 are not shown in all plots in Figure S5. Chromosome 12 is shown in Figure S5 but not explained in the text.

- Line 358-359 - MFA seq experiment cannot be compared with the experiment shown in Figure S1B, as two-colour FACS was not performed under chronic HU treatment.

- Fig1C - to put the statistical analysis on top of the quantification graph.

- Fig 1E, 5D and E – There is no legend at the figure to explain what is being shown.

- Fig. 5B and S4 - to show the chromosome length grid to each plot.

- Line 595 - The amount of DNA should be expressed in μg.

- Line 1072 -1073 - to determine exactly what is shown in Figure 3D. Are the cells shown from chronic (5 hours) or acute (4 hours) treatment?

- To specify the dilution of the antibody used in the Methods section. Some data are missing, e.g. antibody dilution in the paragraph: "S phase detection and cell cycle profile using flow cytometry".

- Sometimes the tagged strain is labelled as mycATR (S2B) and sometimes as 3MATR (S2C) and sometimes as 3xMycATR on the figures. It would be nice if the labelling of the tagged strains (and other strains) were consistent.

- In the text explaining the two-colour FACS analysis (lines 167-169) it is written that it is the mycATRΔC+/- strain shown in S1B, but in the S1B figure it is labelled as mycATRΔC-/-.

- Line 216 - to say how long the cells were grown in the presence of IdU.

- In the text it is stated that the authors used anti-T. brucei γH2A antiserum for Fig. 4A and B, and in these figures the upper panels of the Western blots are labelled as signals detected with anti-myc?

- Figure 4E has very low-resolution quality. In panel 1N0K an example of 2N1K is shown below.

- Fig. 4D - The color code of two grey colors (1N1K and 1N2K) needs to be changed to make these colors distinct. The lines of the bars are too thick and hide the color information.

- Line 372 Are these untreated cells in Figure 5F or 'post chronic HU treatment' cells?

- line 1150 Figure 7A has no figure legend

- Line 1102-1103 - This sentence is probably in the wrong place. In Figure 5A, 10 passages are not shown?

- There are occasional typos or missing or inconsistent information such as

Line 71 DNA lesions(1,2).

Line 185 Leishtag.org).

Line 201 mNG signal. (Figure).

Line 289 ATR results results

Line 488 ‘ATR plays a central role in the cellular response to DNA damage, utilizing its kinase activity is used to regulate numerous repair pathways’

Line 1025 tio

Line 1026 At Figure 1 is written (~6220 – 9624 pb) while in the text ‘the C-terminus of ATR (from 6180 to 9624 bp)’.

Line 1067 Acute treatment is sometimes described as 5 mM hydroxyurea (HU) for 6 hours and sometimes as 5 mM hydroxyurea (HU) for 5 hours.

Line 1076-1078 ‘Percentage of EdU positive mycATR or mycATRΔC-/- cells in each condition was calculated by the number of stained cells by the number of cells present in each sample’

Line 1082 ‘the blort’

Line 1124-1125 rp1 and rp2 are not indicated at Fig 5F

1148 6D bottom or left?

Line 1115 ‘at the right-top’ instead of at the right-bottom

**Have all data underlying the figures and results presented in the manuscript been provided?**

Reviewer #1: Yes

Reviewer #2: None

Reviewer #3: **No: ** Please see my comments in the window above this one.

PLOS authors have the option to publish the peer review history of their article (what does this mean? ). If published, this will include your full peer review and any attached files.

**Do you want your identity to be public for this peer review?** For information about this choice, including consent withdrawal, please see our Privacy Policy .

Reviewer #1: No

Reviewer #2: No

Reviewer #3: No

**Figure resubmission:**
---

## [Decision Letter · Decision Letter 1]

25 Jun 2025

PGENETICS-D-25-00053R1

ATR, a DNA damage kinase, modulates DNA replication timing in Leishmania major

PLOS Genetics

Dear Dr. McCulloch,

Thank you for submitting your manuscript to PLOS Genetics. After careful consideration, we feel that it has merit but does not fully meet PLOS Genetics's publication criteria as it currently stands. Therefore, we invite you to submit a revised version of the manuscript that addresses the points raised during the review process.

Please submit your revised manuscript within 30 days Jul 25 2025 11:59PM. If you will need more time than this to complete your revisions, please reply to this message or contact the journal office at plosgenetics@plos.org. Please include the following items when submitting your revised manuscript:

We look forward to receiving your revised manuscript.

Kind regards,

Dmitry A. Gordenin, Ph.D.

Academic Editor

PLOS Genetics

Geraldine Butler

Section Editor

PLOS Genetics

Aimée Dudley

Editor-in-Chief

PLOS Genetics

Anne Goriely

Editor-in-Chief

PLOS Genetics

**Journal Requirements:**

1) Please make sure to update the legends for Figures 5A and 7A to specify that they were created using BioRender, rather than including this information in the acknowledgments.

**Reviewers' comments:**

Reviewer's Responses to Questions

**Comments to the Authors:**

Reviewer #1: This manuscript was originally reviewed with the recommendation to revise. Upon re-review of the revised manuscript, I find that the minor comments (figure mis-labelling, text corrections, elucidation of the deleted region of ATR in Fig 1, etc.) have mostly been addressed. Note that this has not always been done accurately: as examples:

a) the CNV data (originally Fig S8, now moved to main Fig 7) is labelled D in the Fig 7, but there is no D in the legend. I think it should be C?

b) Low resolution pictures in Fig 4 were raised by several reviewers: these have been replaced, but with pictures that reduce the cell-staining resolution and only slightly improve the DNA resolution.

Most of the more substantive comments in my original review have not been addressed, with the authors respectfully disagreeing, or side-stepping the issue (such as on counterintuitive SNP accumulation, or on the semantics of ‘dormant’ or ‘stochastic’ origins – which, however termed, are poorly detected by MFAseq). Most importantly, some issues have been addressed with added data that only raise further concerns.

Accordingly, the editor should decide whether a somewhat tidied-up but fundamentally largely unchanged manuscript can be published.

To expand on the main outstanding points:

The major suggestion that MFAseq has insufficient resolution (also noted by Reviewer 3) has not been addressed by additional work with DNAscent or combing. This was expected, but as it stands, the text still fails to highlight the very marked shortcomings of the MFAseq technique used alone in this context.

The other major point – that the two ATR-mutant clones c1 and c8 appear to differ greatly in multiple experiments – is only minimally addressed (by an added comment in the text: ‘and randomness in the loci at which they are activated could explain the differences in MFA-seq profiles observed between the mutant clones.’) The concern remains, therefore, that the reported phenotypes are substantially affected by secondary adaptations in different clones. Even more concerningly, where data from 2 technical replicates on the same clones are now added (e.g. Fig S4, this in response to a related concern on this same point, raised by Reviewer 2), it becomes clear that even technical replicates vary hugely, particularly in cl8. The paper hinges on phenotypes of these 2 clones, so their inconsistency (both inter-clone, and even more so, inter-repeat) is a real concern.

Reviewer #2: The authors have responded to most of my concerns. The last part, about SNVs and indels, remains weaker than the rest of the manuscript. While improved from the initial submission, the authors could include the benchmark data they cited in their response as part of the QC for their current analysis. The different patterns of SNVs vs indels are still sensitive to the experimental design (specifically, just 10 generations), and their suggestion that SNVs are less harmful than indels and that a DDR defect may modulate mutation patterns with respect to these different levels of effect, is contested. The authors should present their results more carefully, avoid unsupported interpretations, and note the possible caveats of this experiment.

Reviewer #3: The authors made substantial modifications to the manuscript in order to address the reviewers' comments and clarify certain results. However, they did not complete Supplementary Table S1 as requested. This table was supposed to list all the MFA-seq peaks detected in the wild-type (WT), mycATR and mycATRΔC-/- mutants. In this table, the authors now provide the locations and lengths of the ERR, SSR/SIDER1/SIDER2 and SIDER1/SIDER2. They do not specify which ERRs are shown. I assume these are from the WT strain, since the ERRs from the mycATRΔC⁻/⁻ mutants are not always in these positions (see Figure S4). The MFA-seq results from two replicates often differ significantly (Figure S4). The authors did not specify whether the two replicates presented in Figure S4 were normalised. If so, how can we explain the loss of some of the most intense signals present in one replicate but absent in the other after merging the two replicates?

The authors demonstrate the location of SSR-Oris at EERs. I assume they presume these are the origins of DNA replication. However, they do not explain how they identified SSR-Oris’ locations. Did they use MFA-seq or DNAscent data? Of the DNAscent molecules, how many showed replication cantered on the SSRs, and how many did not? These details must be specified to prevent misinterpretation of the positions of origins.

The authors state that ‘Chromosome 1 was excluded because an amplification in copy number was found in our stationary control sample’. This is interesting, given that the same group previously demonstrated an MFA-seq signal for chromosome 1 (PMID: 26481451). The authors should discuss how the amplification of chromosome 1 can occur in the stationary control sample and explain what this amplification means.

If the authors performed FACS cell sorting of Leishmania for MFA-seq, they should specify this in the Methods section, along with how they did it.

The authors did not provide any proof of DNA replication restart and misinterpreted the data on EdU incorporation (Figure 3E). They demonstrated that EdU was still being incorporated following the acute or chronic HU treatment of mycATR and two mycATRΔC-/- mutants. This could be due to recombination, DNA repair, or replication. The restart of DNA replication can be demonstrated using DNA combing or DNA fiber stretching experiments. Therefore, the authors should remove the following sentence:

Line 277-278 – ‘Together, these data demonstrate that ATR is crucial for efficient DNA replication restart in L. major.’

Minor:

Even in high-resolution, the quality of Figures 3D and 4E is poor.

The line weights in Figure 4F should be reduced to make the borders between the different categories more distinguishable.

It seems that there is an error in the S3 legend. Is it the ssDNA or the EdU signal intensity?

Table S1. - The labelling of the columns is confusing (e.g. column SSR, SIDER1 SIDER2 vs column SIDER1 SIDER2?). A more detailed explanation is required.

The title of Table S2 is 'Table 1'.

**Have all data underlying the figures and results presented in the manuscript been provided?**

Reviewer #1: Yes

Reviewer #2: Yes

Reviewer #3: None

PLOS authors have the option to publish the peer review history of their article (what does this mean? ). If published, this will include your full peer review and any attached files.

**Do you want your identity to be public for this peer review?** For information about this choice, including consent withdrawal, please see our Privacy Policy .

Reviewer #1: No

Reviewer #2: No

Reviewer #3: No

**Figure resubmission:**
---

## [Decision Letter · Decision Letter 2]

30 Jul 2025

PGENETICS-D-25-00053R2

ATR, a DNA damage kinase, modulates DNA replication timing in Leishmania major

PLOS Genetics

Dear Dr. McCulloch,

Thank you for submitting your manuscript to PLOS Genetics. After careful consideration, we feel that it has merit but does not fully meet PLOS Genetics's publication criteria as it currently stands. We note that your submission does not contain numeric data underlying the figures and results presented in the manuscript.  These should be provided as Supplementary part or in other way described in our data availability policy  (also see below). Therefore, we invite you to submit a revised version of the manuscript that addresses this point.

Please submit your revised manuscript within 30 days Aug 29 2025 11:59PM. If you will need more time than this to complete your revisions, please reply to this message or contact the journal office at plosgenetics@plos.org. Please include the following items when submitting your revised manuscript:

We look forward to receiving your revised manuscript.

Kind regards,

Dmitry A. Gordenin, Ph.D.

Academic Editor

PLOS Genetics

Geraldine Butler

Section Editor

PLOS Genetics

Aimée Dudley

Editor-in-Chief

PLOS Genetics

Anne Goriely

Editor-in-Chief

PLOS Genetics

**Reviewers' comments:**

Reviewer's Responses to Questions

Reviewer #2: The authors have partially addressed my remaining concerns.

**Have all data underlying the figures and results presented in the manuscript been provided?**

Reviewer #2: None

PLOS authors have the option to publish the peer review history of their article (what does this mean? ). If published, this will include your full peer review and any attached files.

**Do you want your identity to be public for this peer review?** For information about this choice, including consent withdrawal, please see our Privacy Policy .

Reviewer #2: No

**Figure resubmission:**
---

## [Editor Report · Decision Letter 3]

2 Sep 2025

PGENETICS-D-25-00053R3

ATR, a DNA damage kinase, modulates DNA replication timing in Leishmania major

PLOS Genetics

Dear Dr. McCulloch,

Thank you for submitting your manuscript to PLOS Genetics. After careful consideration, we feel that it has merit but does not fully meet PLOS Genetics's publication criteria as it currently stands. Therefore, we invite you to submit a revised version of the manuscript that addresses the points raised by Editors during the review process (see under Editor's comments below).

Please submit your revised manuscript within 30 days Oct 02 2025 11:59PM. If you will need more time than this to complete your revisions, please reply to this message or contact the journal office at plosgenetics@plos.org. Please include the following items when submitting your revised manuscript:

We look forward to receiving your revised manuscript.

Kind regards,

Dmitry A. Gordenin, Ph.D.

Academic Editor

PLOS Genetics

Geraldine Butler

Section Editor

PLOS Genetics

Aimée Dudley

Editor-in-Chief

PLOS Genetics

Anne Goriely

Editor-in-Chief

PLOS Genetics

**Editor's comments:**

You current supplementary section included into version that we inspected is just a set of images combined into a single pdf file. Please, have all numeric data supporting conclusions from your analyses and graphs and diagrams in your main and supplementary display items presented in spreadsheet format, i.e., excel or tab-delimited text. You current supplementary pdf also includes the image of a computer script, which makes it impossible for verification by others without unnecessary effort of retyping it.  Computer script should be provided as tab-delimited text file. If you decide to re-submit PLOS Office staff and then Editors will examine if reformatted supplementary can be used by members of scientific community to verify your conclusions, compare with data from studies ,and to explore independent hypotheses.

Please, note that PLOS explicitly states in submission guidelines:

For smaller data sets and certain data types, authors may provide their data within supporting information files  accompanying the manuscript. Authors should take care to maximize the accessibility and reusability of the data by selecting a file format from which data can be efficiently extracted (for example, spreadsheets or flat files should be provided rather than PDFs when providing tabulated data).

For more information on how best to provide data, read our policy on data availability . 

**Figure resubmission:**
---

## [Editor Report · Decision Letter 4]

29 Sep 2025

Dear Dr McCulloch,

We are pleased to inform you that your manuscript entitled "ATR, a DNA damage kinase, modulates DNA replication timing in Leishmania major" has been editorially accepted for publication in PLOS Genetics. Congratulations!

Yours sincerely,

Geraldine Butler

Section Editor

PLOS Genetics

Geraldine Butler

Section Editor

PLOS Genetics

Aimée Dudley

Editor-in-Chief

PLOS Genetics

Anne Goriely

Editor-in-Chief

PLOS Genetics

BlueSky: @plos.bsky.social

Comments from the reviewers (if applicable):

**Data Deposition**

http://datadryad.org/submit?journalID=pgenetics&manu=PGENETICS-D-25-00053R4

**Press Queries**

---

## [Editor Report · Acceptance letter]

PGENETICS-D-25-00053R4

ATR, a DNA damage kinase, modulates DNA replication timing in Leishmania major

Dear Dr McCulloch,

We are pleased to inform you that your manuscript entitled "ATR, a DNA damage kinase, modulates DNA replication timing in Leishmania major" has been formally accepted for publication in PLOS Genetics! Your manuscript is now with our production department and you will be notified of the publication date in due course.

With kind regards,

Anita Estes

PLOS Genetics

On behalf of:
